# Observationally Constrained Analysis on the Distribution of Fine and Coarse Mode Nitrate in Global Models

Mingxuan Wu[1], Hailong Wang[1], Zheng Lu[2], Xiaohong Liu[2], Huisheng Bian[3], David Cohen[4], Yan Feng[5], Mian Chin[3], Didier A. Hauglustaine[6], Vlassis A. Karydis[7], Marianne T. Lund[8], Gunnar Myhre[8], Andrea Pozzer[9], Michael Schulz[10], Ragnhild B. Skeie[8], Alexandra P. Tsimpidi[7], Svetlana G. Tsyro[10], Shaocheng Xie[11]

[1]Atmospheric, Climate, & Earth Sciences Division, Pacific Northwest National Laboratory, Richland, WA, USA

[2]Department of Atmospheric Sciences, Texas A & M University, College Station, TX, USA

[3]NASA Goddard Space Flight Center, Greenbelt, Maryland, USA

[4]Australian Nuclear Science and Technology Organisation, Lucas Height, NSW, Australia

[5]Environmental Science Division, Argonne National Laboratory, Argonne, IL, USA

[6]Laboratoire des Sciences du Climat et de l'Environnement (LSCE), UMR8212, CEA-CNRS-UVSQ, Gif-sur-Yvette, France

[7]Institute of Energy and Climate Research: Troposphere (ICE-3), Forschungszentrum Jülich GmbH, Jülich, Germany

[8]Center for International Climate and Environmental Research – Oslo (CICERO), Oslo, Norway

[9]Department of Atmospheric Chemistry, Max Planck Institute for Chemistry, Mainz, Germany

[10]Research and Development Department, Norwegian Meteorological Institute, Oslo, Norway

[11]Lawrence Livermore National Laboratory, Livermore, CA, USA

*Correspondence to*: Mingxuan Wu (mingxuan.wu@pnnl.gov) and Hailong Wang (hailong.wang@pnnl.gov)

**Abstract.** Nitrate plays an important role in the Earth system and air quality. A key challenge in simulating the lifecycle of nitrate aerosol in global models is to accurately represent mass size distribution of nitrate aerosol. In this study, we evaluate the performance of the Energy Exascale Earth System Model version 2 (E3SMv2) and the Community Earth System Model version 2 (CESM2), along with Aerosol Comparisons between Observations and Models (AeroCom) phase III models, in simulating spatial distribution of fine-mode nitrate, mass size distribution of fine- and coarse-mode nitrate, and the gas-aerosol partitioning between nitric acid gas and nitrate, using long-term ground-based observations and measurements from multiple aircraft campaigns. We find that most models underestimate the annual mean $PM_{2.5}$ (particulate matter with diameter less than 2.5 µm) nitrate surface concentration averaged over all sites. The observed nitrate $PM_{2.5}/PM_{10}$ and $PM_1/PM_4$ ratios are influenced by the relative contribution of fine sulfate/organic particles and coarse dust/sea salt particles. Overall, the ground-based observations give an annual mean surface nitrate $PM_{2.5}/PM_{10}$ ratio of 0.7. Most models underestimate the annual mean $PM_{2.5}/PM_{10}$ ratio in all regions. There are large spreads in the modeled nitrate $PM_1/PM_4$ ratios, which span the full range from 0 to 1. Most models underestimate the surface molar ratio of nitrate to total inorganic nitrate averaged across all sites. Our study indicates the importance of gas-aerosol partition parameterization and simulation of dust and sea salt in correctly simulating mass size distribution of nitrate.

## 1 Introduction

Nitrate plays an important role in the Earth's climate and air quality (Boucher et al., 2013; Szopa et al., 2021; Gong et al., 2024). As part of atmospheric aerosols, it can scatter solar radiation (e.g., van Dorland et al., 1997; Adams et al., 2001), change cloud properties by acting as cloud condensation nuclei (CCN) (e.g., Kulmala et al., 1993), and affect atmospheric chemistry (e.g., Bassett and Seinfeld, 1983; Dentener et al., 1996; Liao and Seinfeld, 2005). Despite its important roles, large uncertainties exist in the simulated lifecycle of nitrate aerosol and its radiative forcing from aerosol-radiation interactions (RFari) in global climate models (GCMs) and chemical transport models (CTMs) (e.g., Myhre et al., 2013; Bian et al., 2017; An et al., 2019; Lu et al., 2021; Zaveri et al., 2021; Wu et al., 2022). Global nitrate burdens from nine models participating in the Aerosol Comparisons between Observations and Models (AeroCom) phase III range from 0.03 to 0.43 Tg N (Bian et al., 2017). RFari of nitrate aerosol documented in the Intergovernmental Panel on Climate Change (IPCC) Fifth Assessment Report (AR5) has a wide range of $-0.30$ to $-0.03$ W m$^{-2}$ (1750-2010) (Boucher et al., 2013). Very few studies have assessed nitrate RF from aerosol-cloud interactions (RFaci) (e.g., Xu and Penner, 2012; Lu et al., 2021; Wu et al., 2022). Recent studies using the U.S. DOE's Energy Exascale Earth System Model version 2 (E3SMv2) and NCAR's Community Earth System Model version 2 (CESM2) estimated RFaci of nitrate aerosol to be around $-0.35$ to $-0.22$ W m$^{-2}$ (Lu et al., 2021; Wu et al., 2022). These estimates indicate a substantial impact of nitrate aerosol on the Earth's climate and contribution of nitrate aerosol to the total RF of aerosols.

One key challenge in simulating the lifecycle of nitrate, especially the formation of nitrate aerosol, in GCMs and CTMs is accurately representing the mass size distribution of nitrate aerosol (i.e., the distribution of nitrate mass across particle size range) which often receives less attention and lacks sufficient observational constraints. The dominant pathway of fine- and coarse-mode nitrate formation is different and regionally dependent. Fine-mode ammonium nitrate forms through the thermodynamic interactions between HNO$_3$ and NH$_3$ (excess after fully neutralizing sulfate) (e.g., Bassett and Seinfeld, 1983; Metzger et al., 2002). Coarse-mode nitrate forms mainly through heterogeneous reactions of nitrogen species such as HNO$_3$ and N$_2$O$_5$ on the surface of coarse dust and sea salt particles (e.g., Karydis et al., 2016; Chen et al., 2020; Zhai et al., 2023). Fine-mode nitrate has a more pronounced effect on the CCN number concentration compared to coarse-mode nitrate, thereby significantly affecting RFaci. Bian et al. (2017) found that the coarse-mode fraction of nitrate aerosol from AeroCom phase III models ranges from 0% to >90%. The large spread in modeled mass size distribution of nitrate aerosol can be related to aerosol-chemistry modules having various complexity adopted in GCMs and CTMs to treat nitrate formation as well as model uncertainties in simulating dust and sea salt. All nine AeroCom phase III models use thermodynamic equilibrium models (TEQMs), assuming instantaneous equilibrium between the gas and particle phases, to treat the gas-aerosol partitioning, whereas very few global modeling studies have directly simulated the dynamic gas-particle partitioning of HNO$_3$ (e.g., Feng and Penner, 2007; Xu and Penner, 2012; Lu et al., 2021; Zaveri et al., 2021; Wu et al., 2022). Two of nine AeroCom models do not treat nitrate formation in the coarse mode. Only three models consider nitrate formation on both dust and sea salt particles and adopted the first-order gas-to-particle approximation, instead of

using only TEQMs in the coarse mode, to calculate the rates of heterogeneous reactions of $HNO_3$ onto dust and sea salt particles.

In the past decades, there are only a few regionally focused studies providing observational insights into the mass size distribution of nitrate aerosol, especially the significant contribution of coarse-mode nitrate. They found large relative differences (up to 150%) of nitrate concentrations at co-located desert and marine sites over the U.S. between the Clean Air Status and Trends Network (CASTNET) and the Interagency Monitoring of Protected Visual Environments (IMPROVE) (Ames and Malm, 2001; Sickles and Shadwick, 2008), which suggest significant fractions of coarse-mode nitrate ($PM_{>2.5}$, PM with diameter larger than 2.5 μm) related to heterogeneous reactions on dust and sea salt particles. Long-term measurements of $PM_{2.5}$ and $PM_{10}$ (PM with diameter less than 2.5 and 10 μm, respectively) of nitrate surface concentrations at sites in Japan also show that $PM_{2.5-10}$ (PM with diameter between 2.5 and 10 μm) nitrate accounts for ~50% to ~80% of all $PM_{10}$ nitrate (Khan et al., 2010; Itahashi et al., 2016; Uno et al., 2017; Pan et al., 2018). The same studies also showed that while $PM_{2.5}$ nitrate has a distinct winter-high-summer-low seasonal variation, the seasonal variation of $PM_{2.5-10}$ nitrate is relatively flat. Furthermore, dust induced $PM_{2.5-10}$ nitrate mainly contributes to $PM_{2.5-10}$ nitrate in spring, while sea salt induced $PM_{2.5-10}$ nitrate dominates in other seasons. Although recent studies attempted to compile surface observational data around globe for both fine and coarse aerosols (e.g., Mahowald et al., 2025), there remains a lack of observational constraints on the mass size distribution of nitrate aerosol, especially from a global view.

Previous regional modeling studies showed that including nitrate formation on coarse sea salt and dust particles through heterogenous reactions can significantly shift the mass size distribution of nitrate aerosol (e.g., Chen et al., 2020; Zhai et al., 2023), as it competes for $HNO_3$ with formation of ammonium nitrate on fine particles. Chen et al. (2020) showed that heterogeneous reactions on sea salt shift mass size distribution of nitrate from fine to coarse mode, compared with an experiment turning off sea salt emission using WRF-Chem. The simulated mass size distribution of nitrate agrees well with measurements at Melpitz in Europe, where coarse-mode nitrate ($PM_{>1.2}$, PM with diameter larger than 1.2 μm) accounts for ~20% of total nitrate aerosol in marine air mass in September. Zhai et al. (2023), comparing simulated nitrate concentrations from GEOS-Chem with observations ($PM_1$ and $PM_4$, PM with diameter less than 1 and 4 μm, respectively) from the Korea-United States Air Quality (KORUS-AQ) campaign, found that including heterogeneous reactions on anthropogenic coarse particulate matter, mainly composed of anthropogenic dust, significantly reduces the overestimation of fine-mode nitrate in previous versions of GEOS-Chem.

Finally, only a few global modeling studies have evaluated the spatiotemporal distribution of fine-mode nitrate in GCMs against observations. There remains a lack of comprehensive analysis on the mass size distribution of nitrate aerosol (both fine and coarse mode nitrate) in GCMs. Mezuman et al. (2016) evaluated vertical profiles of $PM_1$ nitrate simulated in the GISS model against measurements from 14 aircraft campaigns and found systematic underestimation of nitrate over US and Europe. Bian et al. (2017) compared vertical profiles of fine-mode nitrate (PM with diameter less than 1 or 2.5 μm) simulated in AeroCom phase III models with measurements from the Arctic Research of the Composition of the Troposphere from Aircraft and Satellites (ARCTAS)

campaign and found the models tend to underestimate fine-mode nitrate below 4 km. Some previous studies compared $PM_{2.5}$ nitrate simulated in GCMs with IMRPOVE observations over the U.S. (e.g., Skeie et al., 2010; Bellouin et al., 2011; Hauglustaine et al., 2014; Zaveri et al., 2021; Lu et al., 2021). More recently, Tsimpidi et al. (2024) evaluated the simulated $PM_1$ and $PM_{2.5}$ nitrate surface concentrations in the EMAC model against $PM_1$ data from field campaigns and $PM_{2.5}$ data from regional networks, respectively, over the past 20 years. This comparison provides a robust basis for assessing model performance in capturing long-term trends and variability of nitrate across key regions in the northern hemisphere.

The goal of this study is to evaluate the performance of E3SMv2, CESM2, along with AeroCom phase III models in simulating (1) spatial distributions of fine-mode nitrate, (2) mass size distribution of fine- and coarse-mode nitrate, and (3) gas-aerosol partitioning between $HNO_3$ and nitrate with long-term ground-based observations and measurements from multiple aircraft campaigns. The paper is organized as follows. Section 2 introduces E3SMv2, CESM2, and AeroCom phase III models with a special focus on the treatment of nitrate formation and describes the model experiments design and observational data. Section 3 evaluates fine-mode nitrate ($PM_1$ and $PM_{2.5}$), mass size distribution of nitrate ($PM_1/PM_4$ and $PM_{2.5}/PM_{10}$ ratios), and gas-aerosol partitioning between $HNO_3$ and nitrate ($NO_3^-/(NO_3^-+HNO_3)$) from model simulations against long-term ground-based observations and measurements from several aircraft campaigns. Discussion and conclusions are presented in section 4.

## 2 Models and Data

### 2.1 Model Description

Among the eight models used in this study, six were part of the AeroCom phase III models that participated in previous intercomparisons relevant to nitrate aerosols (Bian et al., 2017). These AeroCom model experiments, conducted about seven years ago, likely have outdated physical parameterizations and emissions of aerosol and gas species. E3SMv2 and CESM2 are relatively new models that were recently developed to include explicit treatment for nitrate aerosol, so these two models are described in more details here.

### 2.1.1 E3SMv2

We use E3SMv2 (Golaz et al., 2022) with its atmosphere component (EAMv2) and land component (ELMv2). EAMv2 uses almost the same physical package as described in Rasch et al. (2019) and Xie et al. (2018). The Cloud Layers Unified by Binormals (CLUBB) scheme is used to treat boundary layer turbulence, shallow convection, and cloud macrophysics in a unified way (Golaz et al., 2002; Larson et al., 2017). Deep convection is parameterized by the scheme of Zhang and McFarlane (1995) (ZM scheme), with a dynamic Convective Available Potential Energy (dCAPE) trigger (Xie and Zhang, 2000) and an unrestricted air parcel

launch level approach (Wang et al., 2015) as described in Xie et al. (2019). A two-moment cloud microphysics scheme (MG2, Gettelman and Morrison, 2015) is used for large-scale stratiform clouds. Wu et al. (2022) implemented the Model for Simulating Aerosol Interactions and Chemistry (MOSAIC) module (Zaveri et al., 2008) in E3SMv2 and coupled it with the Model for Ozone and Related chemical Tracers gas chemistry (MOZART-4) (Emmons et al., 2010; Tilmes et al., 2015) and an enhanced version of the four-mode version of Modal Aerosol Module (MAM4) (Liu et al., 2016; Wang et al., 2020) following previous model development effort in CESM2 (Lu et al., 2021; Zaveri et al., 2021). MOSAIC is a comprehensive aerosol chemistry module which simulates the dynamic partitioning between semivolatile gases and particles of different sizes in an accurate but computationally efficient way. MOSAIC implemented in EAMv2 replaces the default MAM4 treatment of gas-aerosol exchange between gases, including $H_2SO_4$, $HNO_3$, $NH_3$, HCl, and a single lumped secondary organic aerosol (SOA) precursor, and aerosols. The aqueous chemistry (i.e., occurring with cloud water) is also modified to include reactions of $HNO_3$, $NH_3$, and HCl. In MAM4, aerosol species are assumed to be internally mixed within modes and externally mixed among modes. Wu et al. (2022) added nitrate ($NO_3^-$) and ammonium ($NH_4^+$) aerosol to the Aitken, accumulation, and coarse modes of MAM4. MOSAIC explicitly and independently treats the heterogeneous reactions of $HNO_3$ on particles containing dust (i.e., $CaCO_3$) and/or sea salt (i.e., NaCl) in the Aitken, accumulation, and coarse modes. In EAMv2, calcium ($Ca^{2+}$) and carbonate ($CO_3^{2-}$) aerosols were added in the accumulation and coarse modes with emitted dust mass fractions of 2% and 3%, respectively, for $HNO_3$ reactions on dust following Zaveri et al. (2008). The remaining 95% of the emitted dust in each mode is treated as other inorganic (OIN) in MOSAIC, which does not have chemical reactions with gas and aerosol species. Primary sea salt aerosol in the Aitken, accumulation, and coarse modes is split into three species: sodium ($Na^+$), chloride ($Cl^-$), and sea salt sulfate (ss-$SO_4^{2-}$) with emitted mass fractions of 38.5%, 53.8%, and 7.7%, respectively (Pilson, 2012). We use accommodation coefficients for $HNO_3$, $NH_3$, and HCl of 0.193, 0.092, and 0.1, respectively, following Xu and Penner (2012). We ran E3SMv2 with the spectral-element dynamical core for EAMv2 at 100 km horizontal resolution on a cubed-sphere geometry with 72 vertical layers. Table 1 shows the configuration of gas and aerosol chemistry on nitrate formation in E3SMv2 and other GCMs evaluated in this study.

**2.1.2 CESM2**

We also use CESM2.0 (Danabasoglu et al., 2020) with the Community Atmosphere Model version 6 (CAM6) and the Community Land Model version 5 (CLM5, Lawrence et al., 2019) as the atmosphere and land component, respectively. CAM6 uses similar physical parameterizations as in EAMv2 (i.e., CLUBB, MG2, ZM deep convection, and MAM4 aerosol module). Lu et al. (2021) implemented the MOSAIC in CESM2 and coupled it with MOZART-TS1 gas chemistry (Emmons et al., 2020) and MAM4 aerosol module (Liu et al., 2016) (Table 1). The aerosol speciation, including nitrate ($NO_3^-$), ammonium ($NH_4^+$), dust (i.e., $Ca^{2+}$, $CO_3^{2-}$, and OIN), and sea salt (i.e., $Na^+$ and $Cl^-$), in CAM6 MAM4 for coupling with MOSAIC is the same as in EAMv2 MAM4.

Nitrate and ammonium aerosol are explicitly simulated in the Aitken, accumulation, and coarse modes. However, as shown in Table 1, the geometric standard deviations ($\sigma_g$) in the accumulation and coarse mode of MAM4 are different between CAM6 and EAMv2, which has significant impacts on the lifecycle of dust through dry deposition (Wu et al., 2020). The upper and lower bound of the number median diameter in the three modes are also different between CAM6 and EAMv2. As discussed in Wu et al. (2022), Lu et al. (2021) and Zaveri et al. (2021) used a lower accommodation coefficient for $HNO_3$ ($\leq 0.0011$) in the CESM-MOSAIC, which was measured for $HNO_3$ condensing on pure dust particles and may substantially underestimate the production of nitrate aerosol associated with gas-aerosol partitioning. In this study, we use the same accommodation coefficients of $HNO_3$, $NH_3$, and HCl (0.193, 0.092, and 0.1, respectively) as in E3SMv2-MOSAIC. We ran CESM2.0 with the finite-volume dynamical core for CAM6 at 0.9°×1.25° horizontal resolution with 56 vertical levels.

**2.1.3 AeroCom Phase III models**

As shown in Table 1, we also include six AeroCom Phase III models which simulate nitrate formation in both fine and coarse modes. Three of them (EMAC, OsloCTM2, and OsloCTM3) only use TEQMs for the formation of both fine- and coarse-mode nitrate. Dust (e.g., Ca, Mg, and K) and/or sea salt (e.g., Na and Cl) components are explicitly included in these TEQMs to account for heterogeneous reactions on dust and sea salt particles. Note that OsloCTM2 and OsloCTM3 only consider heterogeneous reactions on sea salt particles. Previous studies showed that supermicron (coarse) nitrate particles take significantly longer, ranging from several hours to days, to reach equilibrium between the gas ($HNO_3$) and particles phase than submicron (fine) particles that equilibrate within minutes (Meng and Seinfeld, 1996; Fridlind and Jacobson, 2000). This equilibrium timescale for supermicron particles exceeds the typical GCM timestep, which is usually less than one hour. TEQMs assume instantaneous equilibrium between gas and particle phases with each timestep, which does not account for the kinetic limitation of coarse particles and therefore may lead to an overestimation of nitrate in the coarse mode. EMAC considers the kinetic limitations in large particles during the process of gas-aerosol partitioning, by using only the amount of the gas-phase species that is able to kinetically condense onto the aerosol phase within the model time step, assuming diffusion limited condensation (Karydis et al., 2016). The other three (EMEP, GMI, and INCA) adopt the first-order loss approximation to calculate nitrate formation in the fine and coarse mode related to heterogeneous reactions on both dust and sea salt particles but still use TEQMs to calculate gas-aerosol partitioning between sulfate, nitrate, ammonium, and gases. As mentioned in Feng et al. (2007), this approach does not explicitly include the equilibrium vapor concentration of species on particle surface as in dynamic mass transfer approach, which depends on relative humidity, temperature, and aerosol chemical composition. Their results show that this hybrid approach produce more coarse-mode nitrate than the dynamical approach (similar to MOSAIC). Unlike the other AeroCom models, GMI reads in a global 3D $HNO_3$ field archived from a previous GMI gas chemistry simulation to calculate nitrate formation. A scaling factor of 0.5 to this $HNO_3$ field

was applied to prevent the overestimation of nitrate aerosol based on a model-observation analysis of the GMI HNO₃. AeroCom

models have different treatments for the size distribution of nitrate, which affects the calculation of nitrate concentration at cut off

size. All AeroCom models except EMEP have relatively coarser horizontal resolutions (2-3 degrees) compared to E3SMv2 and

CESM2.

**Table 1: Nitrate chemical mechanisms and physical properties.**

| Model | Gas-aerosol partitioning method for ammonium nitrate (fine and coarse) | Gas chemistry | DU/SS-nitrate treatment (fine and coarse) | Bins/modes for nitrate | Resolution |
|---|---|---|---|---|---|
| E3SMv2 | DYN (MOSAIC) (Zaveri et al., 2008; Wu et al., 2022) | MOZART-4 (Emmons et al. 2010) | DYN | Aitken, accumulation, and coarse modes ($\sigma_g$=1.6, 1.8, and 1.8) | 1°, 72 |
| CESM2 | DYN (MOSAIC) (Zaveri et al., 2008, 2021; Lu et al., 2021) | MOZART-TS1 (Emmons et al. 2020) | DYN | Aitken, accumulation, and coarse modes ($\sigma_g$=1.6, 1.6, and 1.2) | 1.25°×0.9°, 56 |
| EMAC | TEQM (ISORROPIA-II) (Fountoukis and Nenes, 2007) | MECCA (Sander et al., 2011) | TEQM | nucleation, Aitken, accumulation, and coarse modes | 2.8°×2.8°, 31 |
| EMEP | TEQM (MARS) (Saxena et al., 1986) | EmChem09 (Simpson et al., 2012) | first-order loss | fine and coarse modes ($D_p$=0.33 and 3.0 μm; $\sigma_g$=1.8 and 1.6) | 0.5°×0.5°, 20 |
| GMI | TEQM (RPMARES) (Saxena et al., 1986) | Strahan et al., 2007 | first-order loss | three bins ($D_p$<0.1, 0.1-2.5, >2.5 μm) | 2.5°×2°, 72 |
| INCA | TEQM (INCA) (Hauglustaine et al., 2004) | Hauglustaine et al., 2004; Folberth et al., 2006 | first-order loss | fine and coarse modes | 1.9°×3.75°, 39 |
| OsloCTM2 | TEQM (EQSAM_v03d) (Metzger and Leliveld, 2007) | Berntsen and Isaksen, 1997 | TEQM (only SS) | fine ($D_p$=0.1 μm; $\sigma_g$=2.0) and coarse modes | 2.8°×2.8°, 60 |
| OsloCTM3 | TEQM (EQSAM_v03d) | Berntsen and Isaksen, 1997 | TEQM (only SS) | fine ($D_p$=0.1 μm; $\sigma_g$=2.0) and coarse modes | 2.25°×2.25°, 60 |

Note. DYN: dynamic mass transfer; DU: dust; SS: sea salt.

**2.2 Experiments Design**

We ran both E3SMv2 and CESM2.0 from 2004 to 2014 with 1 year spin-up and the last 10-year results used for analysis. The

horizontal wind components u and v were nudged towards the Modern-Era Retrospective analysis for Research and Applications

Version 2 (MERRA-2, Gelaro et al., 2017) using a relaxation time scale of 6 hours. Monthly mean prescribed historical SST and

sea ice in 2004-2014 from the Hadley Centre Global Sea Ice and Sea Surface Temperature (HadISST) were used. Among the

AeroCom models used in this study, CTMs (i.e., EMEP, GMI, OsloCTM2 and OsloCTM3) were directly driven by reanalysis data,

while GCMs (i.e., EMAC and INCA) were nudged toward reanalysis meteorological data. Both sets of AeroCom models were run

for 2008 following one-year spin-up. In this study, both E3SMv2 and CESM2 use monthly anthropogenic and biomass burning emissions of aerosol and precursor gases specified for the Coupled Model Intercomparison Project phase 6 (CMIP6) (Hosely et al., 2018; van Marle et al., 2017), except for SOA precursors in E3SMv2 that were rescaled from GCM simulation results, as justified in Wang et al. (2020). AeroCom models followed the nitrate experiment protocol (Bian et al., 2017) by using monthly anthropogenic emissions of aerosol and precursor gases from the Hemispheric Transport of Air Pollution (HTAP) v2.2 database (Janssens-Maenhout et al., 2015). Emissions of some VOC gas species not provided by HTAP v2.2 were obtained from CMIP5 RCP85. Biomass burning emissions were from the Global Fire Emissions Database version 3 (GFED3) (van der Werf et al., 2010).

## 2.3 Observations

Table 2 summarizes the ground-based observations used in this study, and Fig. 1 shows the geographic locations of these observational sites. To evaluate simulated $PM_{2.5}$ nitrate surface concentrations (nitrate component of $PM_{2.5}$ concentration), surface nitrate $PM_{2.5}/PM_{10}$ ratios (ratio of $PM_{2.5}$ nitrate to $PM_{10}$ nitrate), and surface molar ratios of particulate nitrate to total nitrate ($NO_3^-/(NO_3^-+HNO_3)$), we use ground-based observations from IMPROVE (Malm et al., 1994) and CASTNET over the U.S. (Fig. 1a), the European Monitoring and Evaluation Programme (EMEP) over Europe (Fig. 1b), and the Acid Deposition Monitoring Network in East Asia (EANET) over East Asia (Fig. 1c). We also use measurements of $PM_{2.5}$ and $PM_{10}$ nitrate surface concentrations at Japanese sites (Fig. 1c) and South African sites (Fig. 1d), obtained from the literature (see Table 2), and measurements of $PM_{2.5}$ nitrate surface concentrations from the Australian Nuclear Science and Technology Organization (ANSTO) Aerosol Sampling Program (ASP) over Australia (Fig. 1e).

There are important differences between $PM_{2.5}$ and $PM_{10}$/total PM (TPM) sampling protocols from those networks and literature, which may make the comparison with model results challenging. We use data from 31 pairs of co-located CASTNET and IMPROVE sites that are separated by less than 0.05° in latitude and longitude for the comparison of surface nitrate $PM_{2.5}/PM_{10}$ ratio. Here we briefly discuss the differences between CASTNET and IMPROVE sampling protocols and their impacts on biases in estimating $PM_{2.5}/PM_{10}$ ratios, as examined by previous studies (Ames and Malm, 2001; Sickles and Shadwick, 2008). 24-h samples of $PM_{2.5}$ nitrate and sulfate are collected on a nylon filter twice a week by IMPROVE Module B sampler with a 2.5 μm cyclone. A carbonate-coated denuder is used, prior to the nylon filter, to remove $HNO_3$. TPM nitrate and sulfate are collected on a Teflon filter weekly by CASTNET filter pack sampler with a non-size selective inlet. $PM_{2.5}$ nitrate measured at IMPROVE sites using Nylon substrate would be overestimated compared with reality if $HNO_3$ were not efficiently removed in the denuder. TPM nitrate measured at CASTNET sites using Teflon substrate may be underestimated compared with reality due to volatilization of ammonium nitrate (Hering and Cass, 1999; Ames and Malm, 2001). EMEP sites have various sampling frequencies for $PM_{2.5}$ nitrate, namely hourly, daily, every three days, four days, or six days, and weekly. The sampling frequencies for $PM_{2.5}$ nitrate are

also different from those for $PM_{10}$ nitrate, mostly daily, at most EMEP sites. $PM_{2.5}$ and $PM_{10}$ nitrate measured at the four Japanese were collected with various sampling frequencies from hourly to twice a week. $PM_{2.5}$ and $PM_{10}$ nitrate reported from Maritz (2019) at four South African sites were collected monthly during 2009-2015. In our analysis, we set surface nitrate $PM_{2.5}/PM_{10}$ ratios to one if the measured $PM_{2.5}$ is larger than the measured $PM_{10}$ (or TPM) after averaging the data.

**Table 2: Summary of the surface observational data used in this study.**

| Surface observation | Quantity | Sites number | Sample frequency | period |
|---|---|---|---|---|
| CASTNET (co-located CSN/IMP) | PM TPM $NO_3^-$ | 31 | Weekly | 2005-2014 |
| CASTNET | TPM $NO_3^-$, $HNO_3$ | 84 | Weekly | 2005-2014 |
| IMPROVE (co-located CSN/IMP) | $PM_{2.5}$ $NO_3^-$ | 31 | Twice a week | 2005-2014 |
| IMPROVE | $PM_{2.5}$ $NO_3^-$ | 162 | Twice a week | 2005-2014 |
| EMEP | $PM_{2.5}$, $PM_{10}$, and TPM $NO_3^-$ | 14 | Hourly, daily, every 3/4/6/7 days | 2005-2014 |
| EMEP | $PM_{10}$ and TPM $NO_3^-$, $HNO_3$ | 14 | Daily | 2005-2014 |
| EANET | TPM $NO_3^-$, $HNO_3$ | 15 | Daily, weekly, bi-weekly | 2005-2014 |
| Komae, Japan (Hayami and Fujita, 2004) | $PM_{2.5}$ and $PM_{10}$ $NO_3^-$ | 1 | Daily | Sep. 1998 – Aug. 2001 |
| Fukue, Japan (Itahashi et al., 2016) | $PM_{2.5}$ and $PM_{10}$ $NO_3^-$ | 1 | Daily, every 3 days | 2002 |
| Yokohama, Japan (Khan et al., 2010) | $PM_{2.5}$ and $PM_{10}$ $NO_3^-$ | 1 | Twice a week | Sep. 2007 – Aug. 2008 |
| Fukuoka, Japan (Uno et al., 2017) | $PM_{2.5}$ and $PM_{10}$ $NO_3^-$ | 1 | Hourly | Aug. 2014 – Oct. 2015 |
| South African sites (Maritz, 2019) | $PM_{2.5}$ and $PM_{10}$ $NO_3^-$ | 4 | Monthly | 2009-2015 |
| Australian sites | $PM_{2.5}$ $NO_3^-$ | 13 | Twice a week | 2005-2014 |

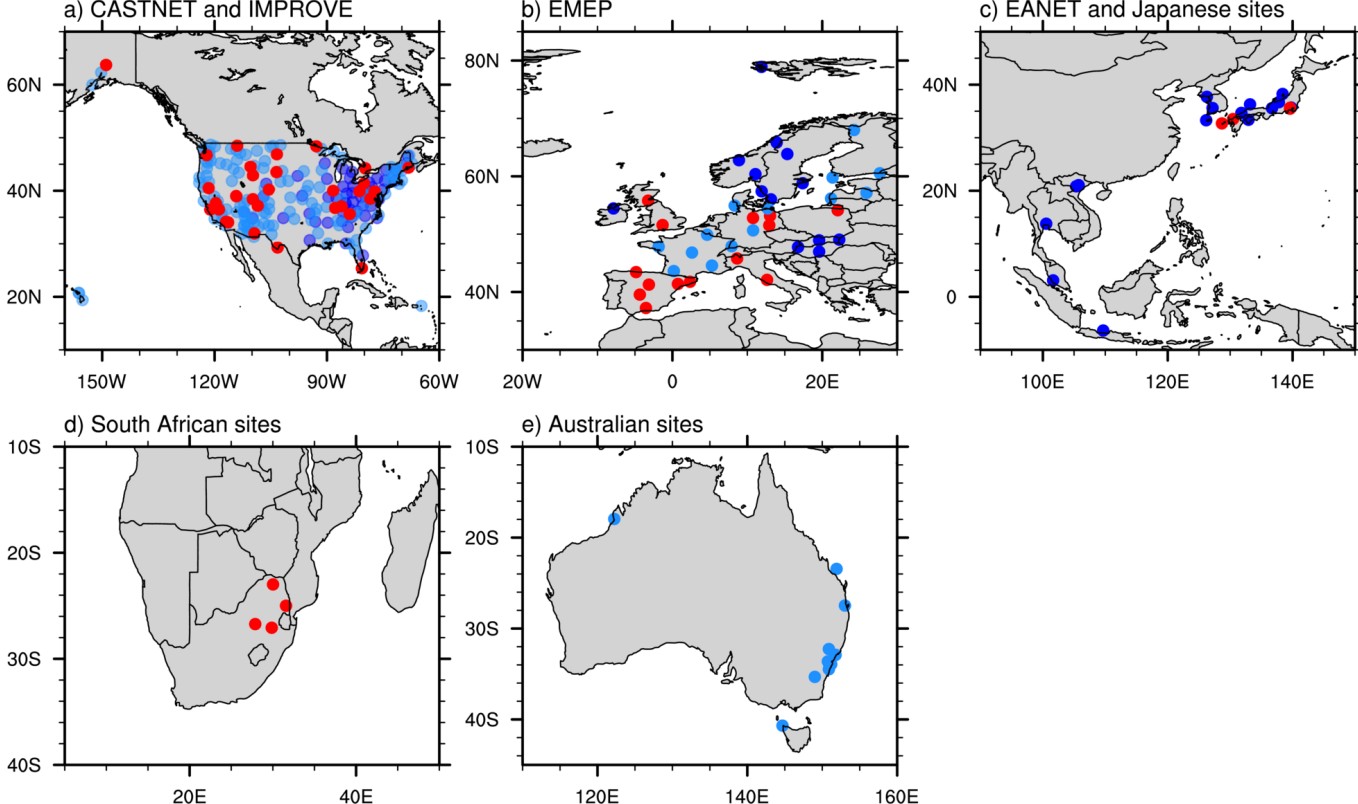

**Figure 1: Illustration of (a) co-located IMPROVE/CASTNET sites (red dots) measuring both PM$_{2.5}$ and TPM nitrate, CASTNET sites (blue dots) measuring both HNO$_3$ and TPM nitrate, and IMPROVE sites (light blue dots) measuring PM$_{2.5}$ nitrate; (b) EMEP sites measuring both PM$_{2.5}$ and PM$_{10}$ nitrate (red dots), EMEP sites measuring both HNO$_3$ and TPM nitrate (blue dots), and EMEP sites measuring PM$_{2.5}$ nitrate (light blue dots); (c) Japanese sites (red dots) measuring both PM$_{2.5}$ and PM$_{10}$ nitrate and EANET sites (blue dots) measuring both HNO$_3$ and TPM nitrate (blue dots); (d) South African sites (red dots) measuring both PM$_{2.5}$ and PM$_{10}$ nitrate; and (e) ANSTO ASP sits (light blue dots) measuring PM$_{2.5}$ nitrate.**

Unlike ground-based observations, only PM$_1$ and PM$_4$ nitrate are measured in aircraft campaigns. In this study, we also compare modeled vertical profiles of fine-mode nitrate aerosol and nitrate PM$_1$/PM$_4$ ratio with measurements of PM$_1$ nitrate from the aerosol mass spectrometer (AMS) and measurements of PM$_4$ nitrate from the Soluble Acidic Gases and Aerosol (SAGA) filters during aircraft campaigns, including ARCTAS (Jacob et al., 2010), the Deep Convective Clouds and Chemistry (DC3) (Barth et al., 2015), the Studies of Emissions and Atmospheric Composition, Clouds, and Climate Coupling by Regional Surveys (SEAC[4]RS) (Toon et al., 2016), the Wintertime Investigation of Transport, Emissions, and Reactivity (WINTER), the Korea-United States Air Quality (KORUS-AQ) (Crawford et al., 2021), and the Atmospheric Tomography Mission (ATom) (Thompson et al., 2021). Figure 2 shows the flight tracks from those campaigns. We use the corresponding monthly mean model results for the aircraft campaigns operated during the simulation period. Note that all aircraft campaigns except for ARCTAS are outside the model simulation year (2008) of AeroCom phase III models. For comparisons with WINTER, KORUS-AQ, and ATom campaigns that are not within the simulation period (2005-2014) of E3SMv2 and CESM2, we use the ten-year average of the corresponding month from the two models. The

model biases in fine-mode nitrate and nitrate $PM_1/PM_4$ ratio, when compared to aircraft measurements, should be interpreted with caution due to discrepancies, particularly in the anthropogenic, biomass burning, and dust aerosol emissions, between the simulation period and the observation period. Those differences in emissions may have stronger impact on the simulated mass size distribution of nitrate in the middle to upper troposphere than in the lower troposphere over remote oceans (e.g., ATom), as sea salt is dominant in the marine boundary layer and fine sulfate/carbonaceous aerosols and coarse dust aerosols are dominant in the middle to upper troposphere (Thompson et al., 2021). Model results are interpolated along the flight tracks based on monthly mean output. We divide ATom observations and model results into eight sectors. Two cutoff sizes, $d_p = 1$ μm and $d_p = 4$ μm, are applied to modeled profiles of aerosols from E3SMv2, CESM2, and EMEP for comparison with measurements from AMS and SAGA filters (Guo et al., 2021; McNaughton et al., 2007), respectively. We set nitrate $PM_1/PM_4$ ratios to one if measured values from AMS is larger than those from SAGA filters after averaging the data. Due to data availability, we use fine-mode nitrate reported by OsloCTM2 and OlsoCTM3, $PM_{2.5}$ nitrate reported by EMAC and INCA, and $PM_1$ nitrate reported by GMI and EMEP in the comparison with AMS measurements from aircraft field campaigns.

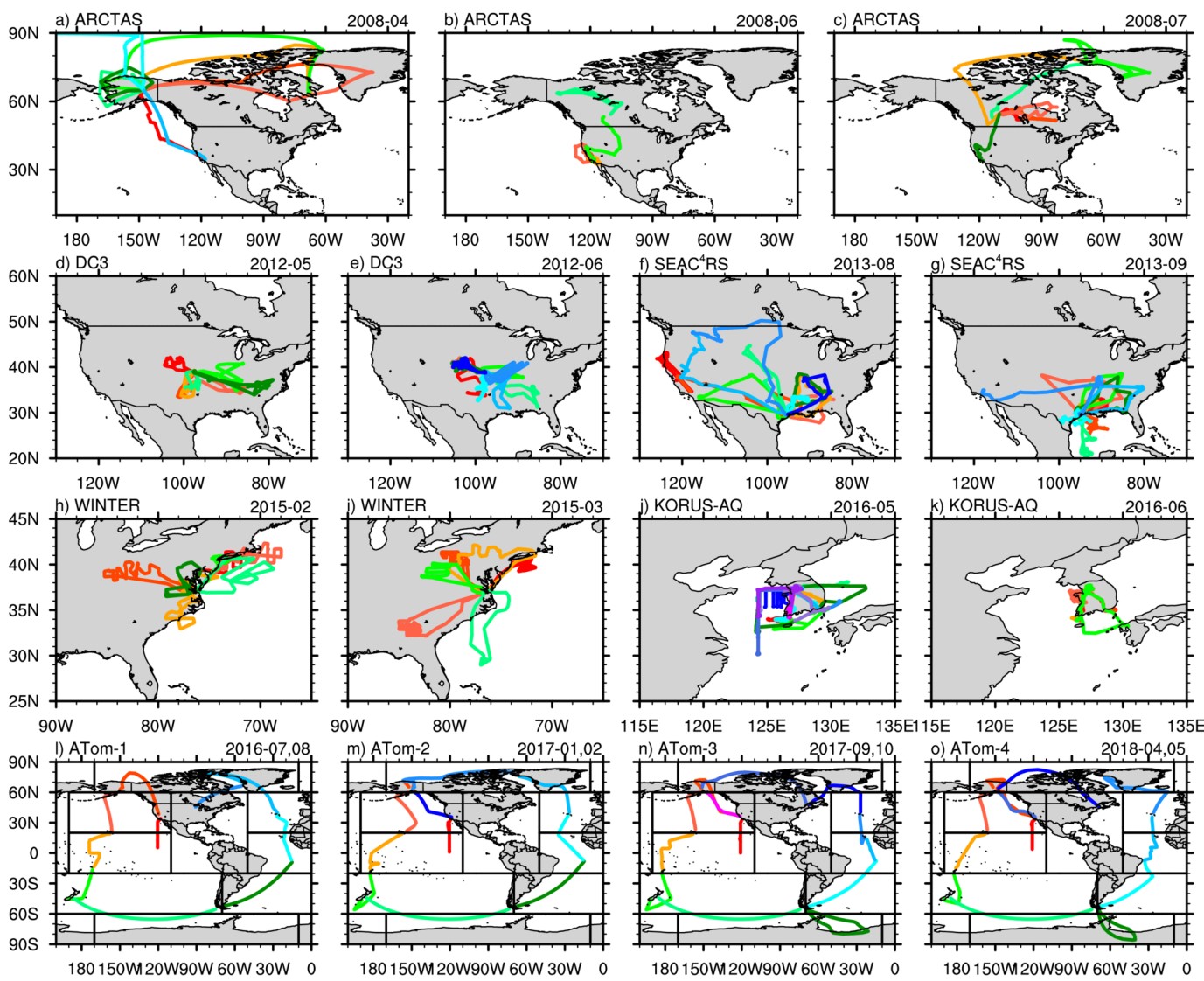

**Figure 2: Flight tracks of ARCTAS, DC3, SEAC⁴RS, WINTER, KORUS-AQ, and ATom campaigns. Different colors in each panel represent different flight days during the month(s) noted at the upper-right corner. Black boxes in the bottom-row panels mark the regions used for the average of observations and model results along Atom flight tracks. The latitudes and longitudes of these regions are (60ºN-90ºN, 170ºW-10ºW), (20ºN-60ºN, 170ºE-110ºW), (20ºN-60ºN, 50ºW-0º), (20ºS-20ºN, 170ºE-110ºW), (20ºS-20ºN, 50ºW-0º), (60ºS-20ºS,**

**160ºE-70ºW), (60ºS-20ºS, 70ºW-0º), and (60ºS-90ºS, 170ºW-10ºW), respectively.**

## 3 Results

As shown in Table 3, there is a large spread in the modeled global annual mean $PM_{2.5}/PM_{10}$ nitrate burden ratios from E3SMv2, CESM2, and AeroCom phase III models, ranging from 0.03 (OsloCTM3) to 0.58 (INCA). The $PM_{2.5}/PM_{10}$ nitrate burden ratios

from OlsoCTM2 and OsloCTM3 are lower than those from all other models, likely because the two models use only TEQMs for the formation of coarse-mode nitrate on sea salt particles. E3SMv2 has notably larger $PM_{2.5}/PM_{10}$ nitrate burden ratio than CESM2,

although both models use MOSAIC for gas-aerosol partitioning. It is partially due to that CESM2 has larger sea salt burden and longer lifetime for sea salt and dust (Table S1). There is a large spread in modeled global annual mean emission and burden of dust and sea salt (see Table S1), which can contribute to the large spread in modeled $PM_{2.5}/PM_{10}$ nitrate burden ratio. There is also a large spread in the modeled global annual mean $NO_3^-/(NO_3^- + HNO_3)$ tropospheric burden ratio, ranging from 0.13 (GMI) to 0.59 (EMEP). We can see that a relatively high or low $HNO_3$ tropospheric burden does not always correlate with a high or low nitrate burden in the models, such as GMI and EMAC. The global annual mean $NO_3^-/(NO_3^- + HNO_3)$ tropospheric burden ratio for GMI would be even lower, considering that GMI scales a global 3D $HNO_3$ field from previous gas chemistry simulation by 0.5 to calculate nitrate formation. We also notice that models having relatively low $PM_{2.5}/PM_{10}$ nitrate burden ratios tend to have relatively high $NO_3^-/(NO_3^- + HNO_3)$ tropospheric burden ratios.

Table 3: Global annual mean of $PM_{2.5}$ nitrate burden, total nitrate burden, $PM_{2.5}/PM_{10}$ nitrate burden ratio, tropospheric $HNO_3$ burden (below 100 hPa), and molar ratio of tropospheric particulate nitrate burden.

| Model | $PM_{2.5}$ $NO_3^-$ burden (Tg N) | $NO_3^-$ burden (Tg N) | $PM_{2.5}/PM_{10}$ $NO_3^-$ burden ratio (mol mol$^{-1}$) | $HNO_3$ trop burden (Tg N) | $NO_3^-/(NO_3^- + HNO_3)$ trop burden ratio (mol mol$^{-1}$) |
|---|---|---|---|---|---|
| E3SMv2 | 0.085 | 0.29 | 0.34 | 0.33 | 0.47 |
| CESM2 | 0.067 | 0.30 | 0.22 | 0.47 | 0.37 |
| EMAC | 0.072 | 0.16 | 0.46 | 0.69 | 0.19 |
| EMEP | 0.067 | 0.22 | 0.34 | 0.15 | 0.59 |
| GMI | 0.029 | 0.059 | 0.49 | 0.40 | 0.13 |
| INCA | 0.10 | 0.18 | 0.58 | 0.33 | 0.35 |
| OsloCTM2 | 0.021 | 0.14 | 0.15 | 0.29 | 0.33 |
| OsloCTM3 | 0.013 | 0.42 | 0.03 | 0.51 | 0.46 |

Note. E3SMv2 and CESM2 simulation period is 2005-2014, while the simulation period for AeroCom models is 2008.

## 3.1 Surface Concentrations and Vertical Profiles of Fine-mode Nitrate

We first evaluate the modeled $PM_{2.5}$ nitrate surface concentrations from E3SMv2, CESM2, and AeroCom phase III models against ground-based observations from IMPROVE over the U.S., EMEP over Europe, the South African sites, and ANSTO ASP sites over Australia (Figure 3). In general, both E3SMv2 and CESM2 overestimate the annual mean $PM_{2.5}$ nitrate surface concentration averaged across all sites, while all AeroCom models underestimate the fine-mode nitrate. The high model biases of $PM_{2.5}$ nitrate surface concentration in E3SMv2, CESM2, and EMAC at IMPROVE sites are consistent with their high model biases of total nitrate aerosol reported in previous studies (Bian et al., 2017; Zaveri et al., 2021; Lu et al., 2021; Wu et al., 2022), which is due to the positive model bias of $HNO_3$. GMI, OsloCTM2, and OsloCTM3 exhibit low model biases for $PM_{2.5}$ nitrate at IMPROVE sites but show high model biases for total nitrate aerosol at CASTNET sites, as reported in Bian et al. (2017). The three models have

much larger negative biases in PM2.5 nitrate than in total nitrate aerosol at EMEP sites, as reported in Bian et al. (2017). All models

significantly underestimate PM2.5 nitrate at the Australian sites except for CESM2. Wu et al. (2020) showed that CESM2 significantly overestimate dust optical depth and extinction profiles over Australia due to an overestimation of local dust emission. CESM2 reduces the geometric standard deviations of particle size distribution in the MAM4 accumulation and coarse modes compared to those in E3SMv2, which reduces dry deposition of dust and sea salt and, consequently, affects their lifecycle. These changes inadvertently increase the nitrate formation and result in the good agreement of PM2.5 nitrate over Australia.

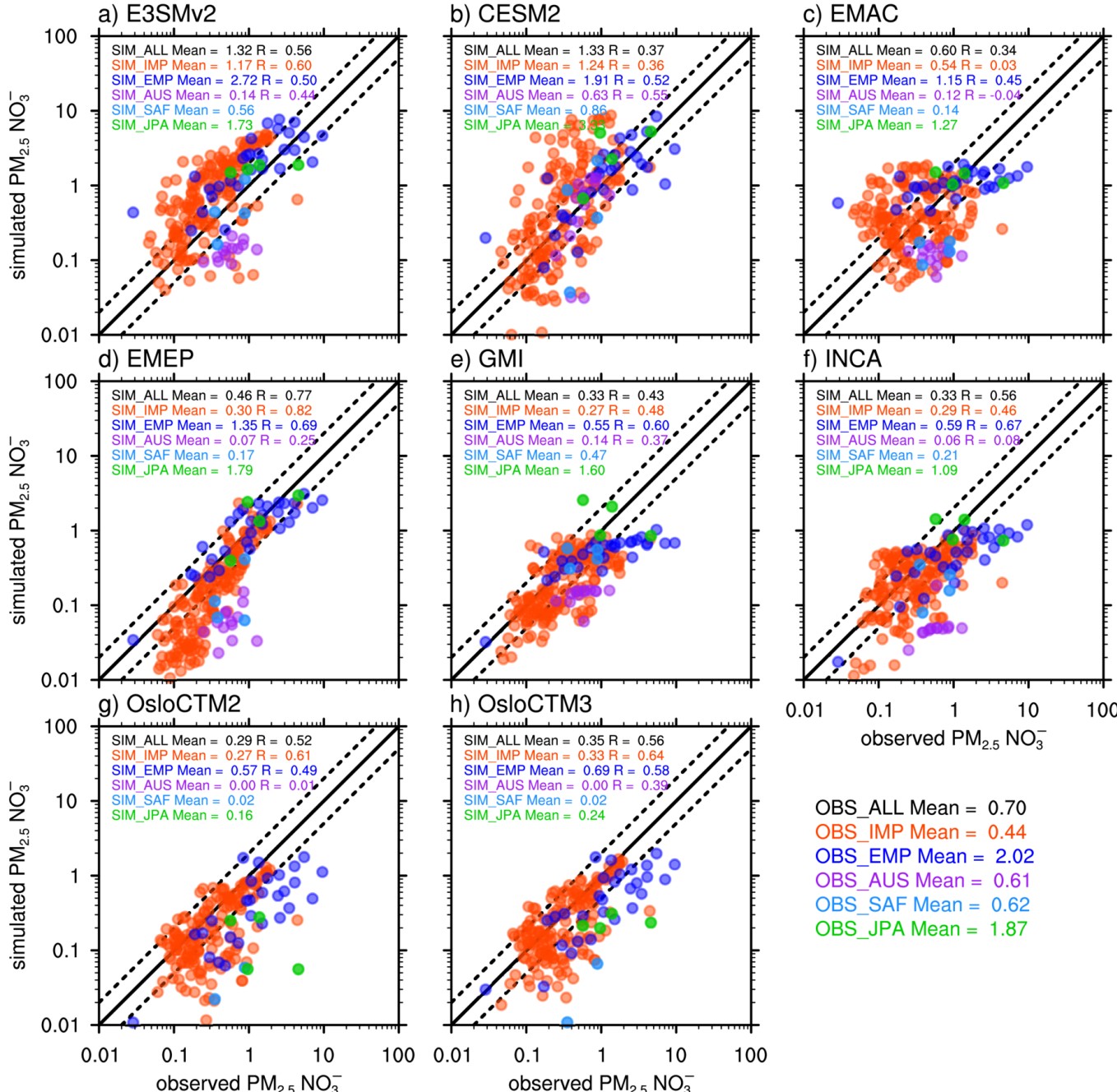

**Figure 3: Scatter plots of modeled annual mean PM2.5 nitrate surface concentrations (μg m⁻³) compared to observations from IMPROVE**

**(IMP, red dots), EMEP (EMP, deep blue dots), the South African sites (SAF, purple dots), the Australian sites (AUS, light blue dots), and the Japanese sites (JPA, light green dots). Model names are noted at the top of individual panels. The numbers in each panel are mean concentrations and correlation coefficients at the corresponding sites (colors). Solid lines represent 1:1 comparison. Dashed lines represent a factor of 2 bias.**

Figure 4 compares the modeled vertical profiles of fine-mode nitrate from E3SMv2, CESM2, and AeroCom phase III models with AMS measurements ($PM_1$) from ARCTAS, DC3, $SEAC^4RS$, WINTER, and KORUS-AQ campaigns. KORUS-AQ has the highest observed $PM_1$ nitrate concentrations (1 to 10 $\mu g\ m^{-3}$) among all campaigns in the lower troposphere (below 800 hPa), as it was operated in Korea and adjacent oceanic regions where nitrate concentrations are generally high. $PM_1$ nitrate was found to be relatively higher (larger than 10 $\mu g\ m^{-3}$) below 1.5 km during the transport period (May 25-31), when there was a significant haze development, and during an atmospheric blocking pattern period (June 01-07) than at other times in May (Park et al., 2021). WINTER, DC3, and $SEAC^4RS$ were all operated in the U.S. and adjacent oceanic regions. We can see that WINTER has higher observed $PM_1$ nitrate concentrations than DC3 and $SEAC^4RS$ in the lower troposphere, as it was operated in February and March when nitrate surface concentrations are seasonally high. The observed large spike of $PM_1$ nitrate around 600 hPa during $SEAC^4RS$ in Fig. 4f was mostly caused by wildfires in the western U.S. (Toon et al., 2016). The observed $PM_1$ nitrate from WINTER strongly decreases from surface to 600 hPa, which is quite different from the vertical variations in the other two campaigns. The observed spikes at 600 hPa in Fig. 4a and at 900 hPa in Fig. 4c may be caused by fire plumes from Siberia, California, and Saskatchewan (Jacob et al., 2010). In general, there are large spreads in the modeled fine-mode nitrate concentrations, which can span three orders of magnitude in some cases. Most AeroCom models underestimate $PM_1$ nitrate concentrations below 600 hPa in ARCTAS, DC3, $SEAC^4RS$, and KORUS-AQ, which were operated between April and September. OsloCTM2 and OsloCTM3 have much lower fine-mode nitrate concentrations and quite different vertical variations of fine-mode nitrate compared to other models and observations.

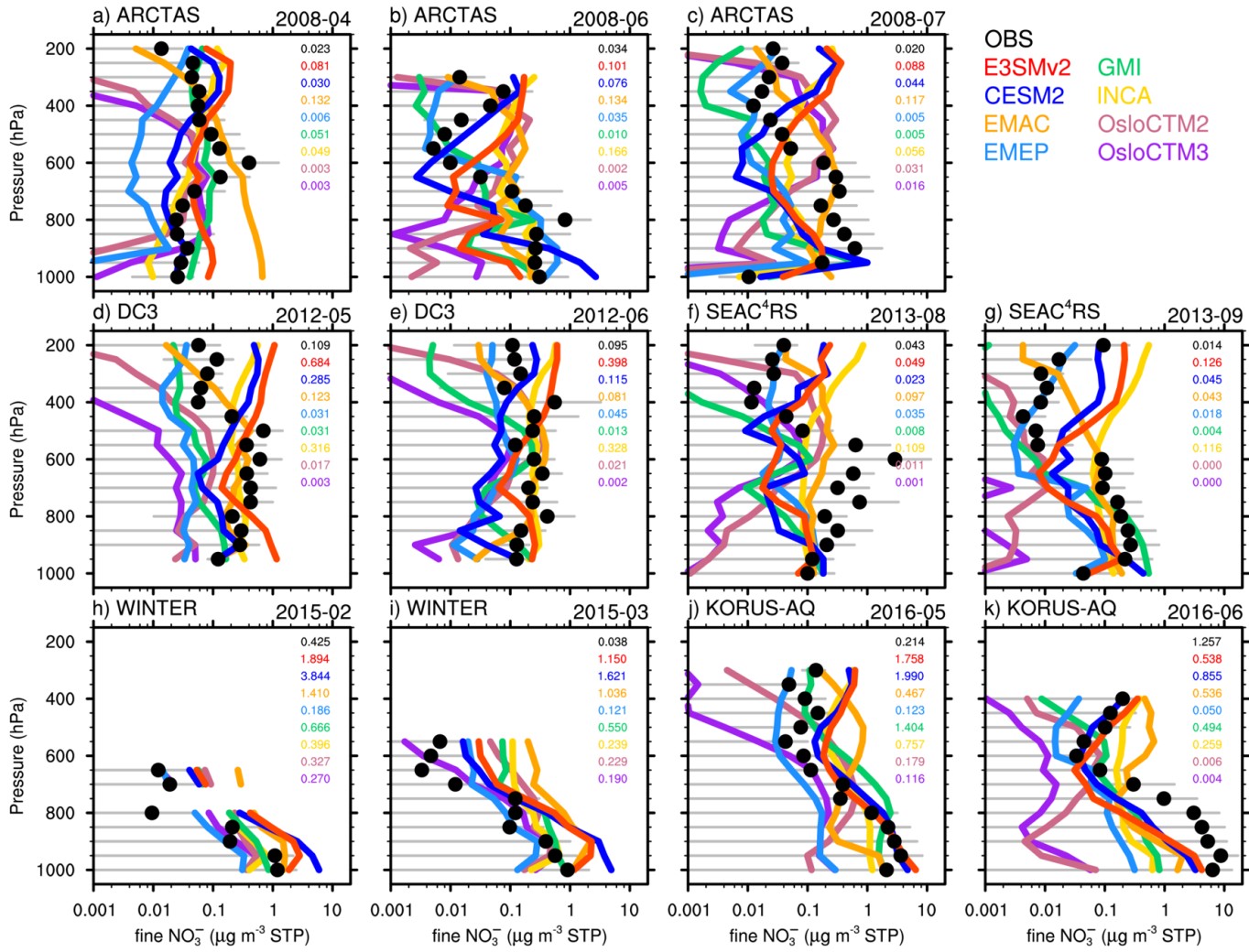

**Figure 4: Vertical profiles of fine-mode nitrate concentrations (μg m⁻³ in STP) from model simulations (colored lines) and PM₁ nitrate concentrations from five aircraft campaigns: (a-c) ARCTAS, (d-e) DC3, (f-g) SEAC⁴RS, (h-i) WINTER, and (j-k) KORUS-AQ. Black dots denote mean values of observations with one standard deviation on each side marked in grey lines. The numbers in each panel are median concentrations.**

Figure 5 compares the modeled vertical profiles of fine-mode nitrate from E3SMv2, CESM2, and AeroCom phase III models with AMS measurements (PM₁) from the ATom campaign. Observed PM₁ nitrate concentrations were between 0.01 and 0.1 μg m⁻³, which are generally lower than the five campaigns in Fig. 4. Similarly, there are large spreads in the modeled fine-mode nitrate concentrations, which can span two orders of magnitude in some cases. EMAC and INCA have significant positive biases in the fine-mode nitrate (at least 3 times larger than observations in most regions), indicating a possible overestimation of PM₁ nitrate. However, it should be noted that the comparison was made using PM₂.₅ nitrate data from these two models instead of PM₁. Fine-mode nitrate from OsloCTM2 and OsloCTM3 has significant negative biases in all regions. Both E3SMv2 and CESM2 show much stronger vertical variations of fine-mode nitrate than all AeroCom models, especially EMAC, EMEP, GMI, and INCA, and

observations. This may be related to the different treatments of CCN activation, wet scavenging, and vertical transport in these two groups of models.

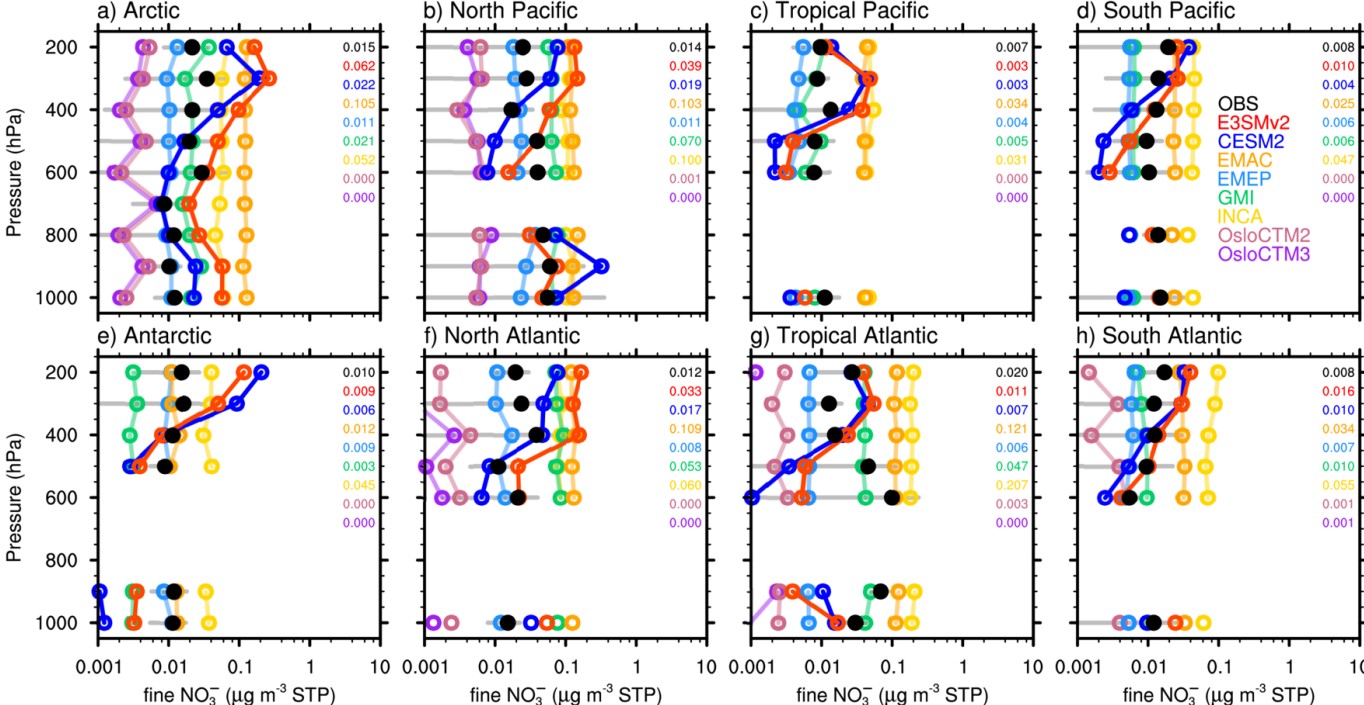

**Figure 5: Vertical profiles of fine-mode nitrate concentrations (µg m⁻³ in STP) from model simulations (colored dots) and PM₁ nitrate**
**concentrations from ATom 1-4 campaigns (with black dots for mean values and grey lines marking one standard deviation of observations). The numbers in each panel are median concentrations.**

**3.2 Mass Size Distribution of Fine- and Coarse-mode Nitrate**

Figure 6 evaluates the modeled surface nitrate $PM_{2.5}/PM_{10}$ ratios from E3SMv2, CESM2, and AeroCom phase III models against
ground-based observations from the co-located IMPROVE/CASTNET sites over the U.S., EMEP sites over Europe, the South African sites, and the Japanese sites (Fig. 1). Overall, the ground-based observations give an annual mean surface nitrate $PM_{2.5}/PM_{10}$ ratio of 0.70. The annual mean $PM_{2.5}/PM_{10}$ ratios in the four regions are 0.82, 0.71, 0.56, and 0.57 respectively. Only a few sites are dominated by coarse-mode nitrate with $PM_{2.5}/PM_{10}$ ratios below 0.5, while most sites are dominated by fine-mode nitrate. The annual mean surface nitrate $PM_{2.5}/PM_{10}$ ratio (0.70), averaged across all sites, is likely higher than the global annual
mean, because the latter includes contributions from vast marine and dusty areas where coarse-mode nitrate is dominant. Most models (CESM2, EMAC, GMI, OsloCTM2 and OsloCTM3) have negative biases in all four regions. Although E3SMv2, EMEP, and INCA slightly overestimate the annual mean surface nitrate $PM_{2.5}/PM_{10}$ ratio, they still have negative biases for considerable

number of sites, especially for co-located IMPROVE/CASTNET sites. EMAC, OsloCTM2, and OsloCTM3, which use only

TEQMs for the formation of coarse-mode nitrate, have larger biases than the other three AeroCom models that use first-order loss

to calculate nitrate formation in the coarse mode for heterogenous reactions on dust and sea salt. E3SMv2 with MOSAIC tends to

agree with observations reasonably well, while CESM2 with MOSAIC have negative biases in all four regions. We also see that

most models have consistent positive or negative biases in all four regions, while EMEP and INCA have different signs of bias

among the regions. Note that the models' performance in simulating surface nitrate $PM_{2.5}/PM_{10}$ ratio does not necessarily align

with global annual mean $PM_{2.5}/PM_{10}$ nitrate burden ratio (Table 3).

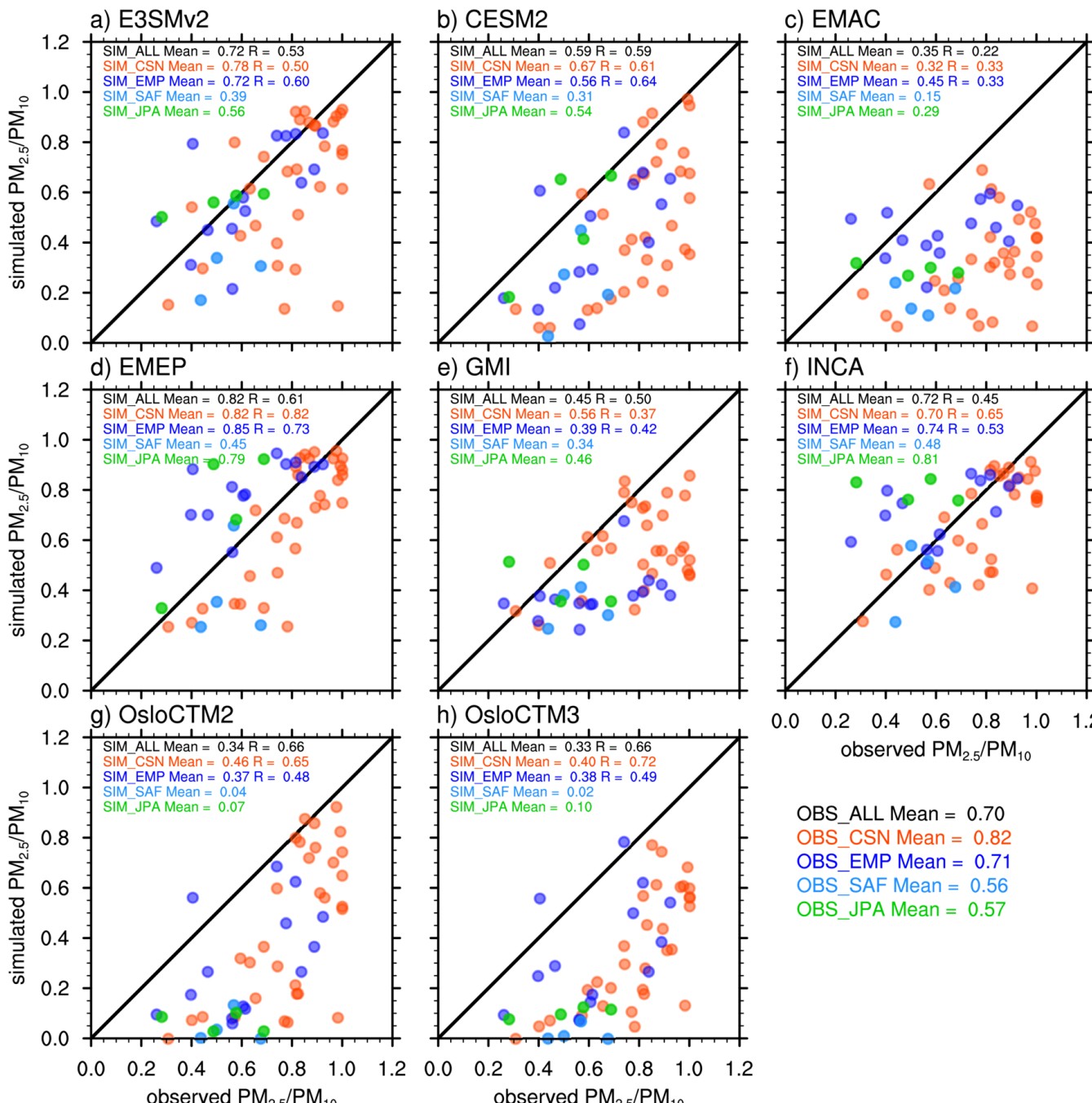

**Figure 6: Scatter plots of modeled annual mean surface nitrate PM$_{2.5}$/PM$_{10}$ ratios compared to observations from the co-located IMPROVE/CASTNET sites (CSN, red dots), EMEP (EMP, deep blue dots), the South African sites (SAF, light blue dots), and the Japanese sites (JPA, green dots). The numbers in each panel are mean ratios and correlation coefficients at the corresponding sites. Solid lines represent 1:1 comparison. The mean surface PM$_{2.5}$/PM$_{10}$ ratio is calculated as mean PM$_{2.5}$ divided by mean PM$_{10}$.**

Figures 7 and 8 show the seasonal variations of modeled surface nitrate PM$_{2.5}$/PM$_{10}$ ratio in comparison with observations at selected co-located IMPROVE/CASTNET sites. In Figure 7, the nitrate surface concentrations at Everglades NP and Virgin Island (Figs. 7a and 7b) are dominated by coarse-mode nitrate due to heterogeneous reactions on sea salt from adjacent ocean and dust

transported from North Africa (see Figs. S1 and S2 for chlorine and $PM_{2.5}$ dust surface concentrations). The measured $PM_{2.5}/PM_{10}$ ratios at the two sites are around 0.3 and 0.2, respectively, with small fluctuations. The higher $PM_{2.5}/PM_{10}$ ratios observed at the two sites from May to August may be attributed to a relatively larger contribution of dust compared with sea salt during this period (see Figs S1 and S2). The nitrate concentrations at Acadia NP (Fig. 7c) are contributed by both coarse-mode nitrate due to heterogeneous reactions on sea salt and fine-mode nitrate. The measured $PM_{2.5}/PM_{10}$ ratios reach their maximum in June, which is consistent with relatively low sea salt concentrations (Fig. S1) and high sulfate concentrations (Fig. S3) in summer. The nitrate concentrations at Big Bend NP, Chiricahua, and Canyonlands NP (Figs. 7d-7f) are dominated by coarse-mode nitrate due to heterogeneous reactions on local dust (see Fig. S2). The measured $PM_{2.5}/PM_{10}$ ratios at the three sites are higher than those at Everglades NP and Virgin Island but below 0.6 during most time of the year. The measured high $PM_{2.5}/PM_{10}$ ratios at Canyonlands NP during winter may be caused by wildfire or air pollution.

OsloCTM2 and OsloCTM3 significantly underestimate the $PM_{2.5}/PM_{10}$ ratios at all six sites with near-zero values in most months. EMEP, GMI, and INCA agree with the observations at Everglades NP and Virgin Island better than other models. All models except for E3SMv2 and EMAC fail to capture the high $PM_{2.5}/PM_{10}$ ratios at Acadia NP in June and July but give high values in winter instead. E3SMv2, EMEP, GMI, and INCA tend to have stronger seasonal variations than observations at Big Bend NP, Chiricahua, and Canyonlands NP. The model biases in $PM_{2.5}/PM_{10}$ ratio at these six sites can be partially attributed to their model biases in simulating dust and sea salt. CESM2 significantly overestimate chlorine aerosol surface concentrations at the six sites, while E3SMv2 has much better agreement with the observations (Fig. S1). This difference in chlorine aerosol can be attributed to several factors, including the reduced geometric standard deviations of particle size distribution in the CESM2 MAM4 accumulation and coarse modes compared to those in E3SMv2 (Wu et al., 2020), the numerical coupling of aerosol emissions, dry deposition, and turbulent mixing in E3SM (Wan et al., 2024), and other differences in cloud and convection parameterizations. CESM2 significantly underestimate $PM_{2.5}$ dust concentrations at the six sites, while E3SMv2 has larger dust concentrations and agree with the observations better (Fig. S2). This difference in dust aerosol can be attributed to the tuned source function for dust emission in CESM2 compared with E3SMv2 (Wu et al., 2020). Note that most AeroCom models underestimate sulfate surface concentrations (Fig. S3). In Figure 8, all six co-located sites are dominated by fine-mode nitrate, mostly ammonium nitrate, as the measured $PM_{2.5}/PM_{10}$ ratios are around or above 0.8. Many models produce much stronger seasonal variations (summer-low-winter-high) of surface nitrate $PM_{2.5}/PM_{10}$ ratio than observations and have large negative biases during May to September. This might be related to that some models fail to capture the seasonal variations (summer-high-winter-low) of sulfate (Fig. S4) and/or model representation of wet deposition. Similarly, CESM2 overestimate chlorine aerosol surface concentrations but underestimate $PM_{2.5}$ dust surface concentrations at the six sites, while E3SMv2 has better agreement with observations (Figs. S5 and S6).

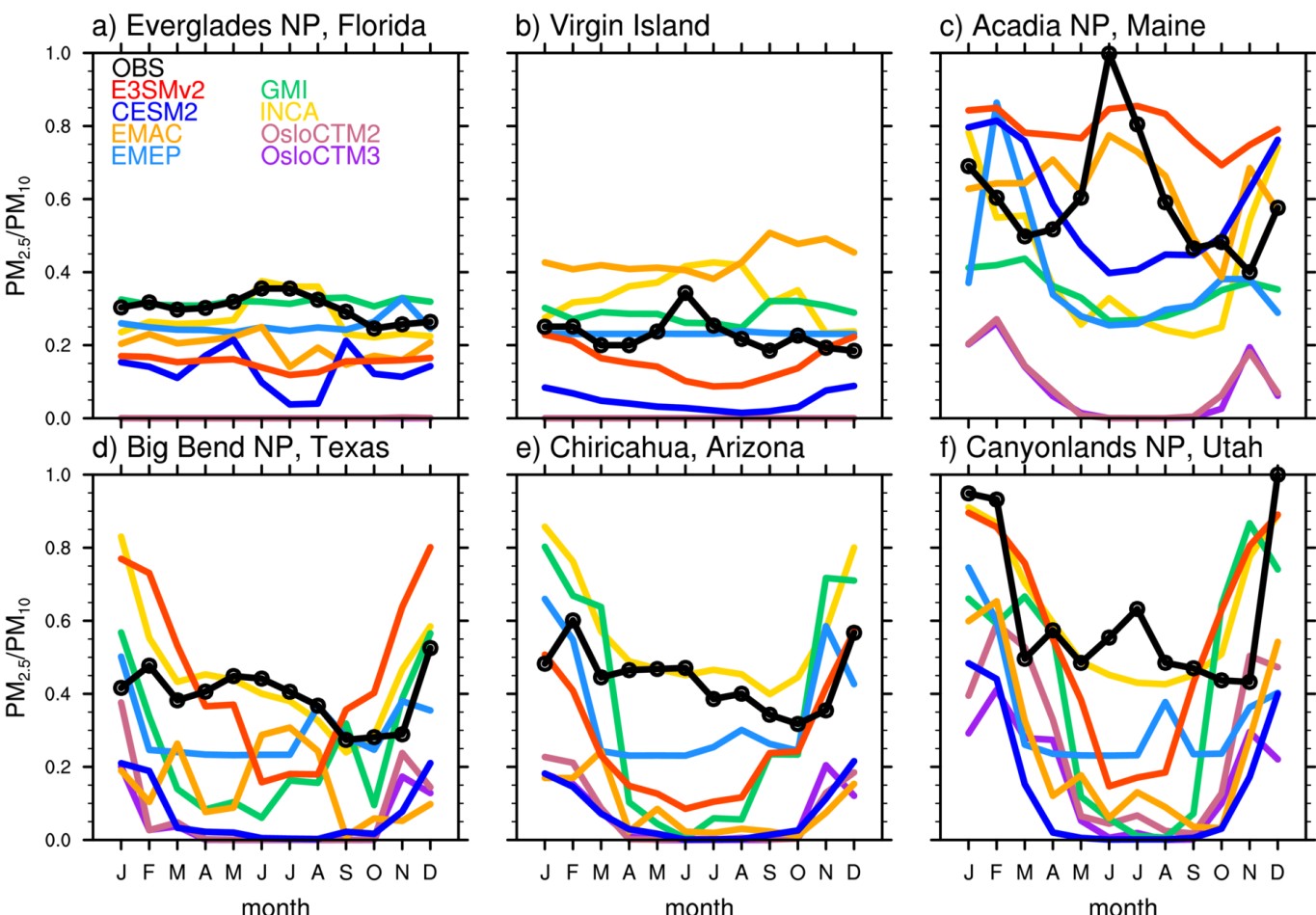

**Figure 7: Seasonal variations of simulated (color lines) and observed (black lines and circles) surface nitrate PM$_{2.5}$/PM$_{10}$ ratio at six co-located IMPROVE/CASTNET sites dominated by coarse-mode nitrate: (a) Everglades NP (25.39ºN, 80.68ºW), (b) Virgin Island (18.34ºN, 64.80ºW), (c) Acadia NP (44.38ºN, 68.26ºW), (d) Big Bend NP (29.30ºN, 103.18ºW), (e) Chiricahua (32.01ºN, 109.39ºW), and (f) Canyonlands NP (38.46ºN, 109.82ºW).**

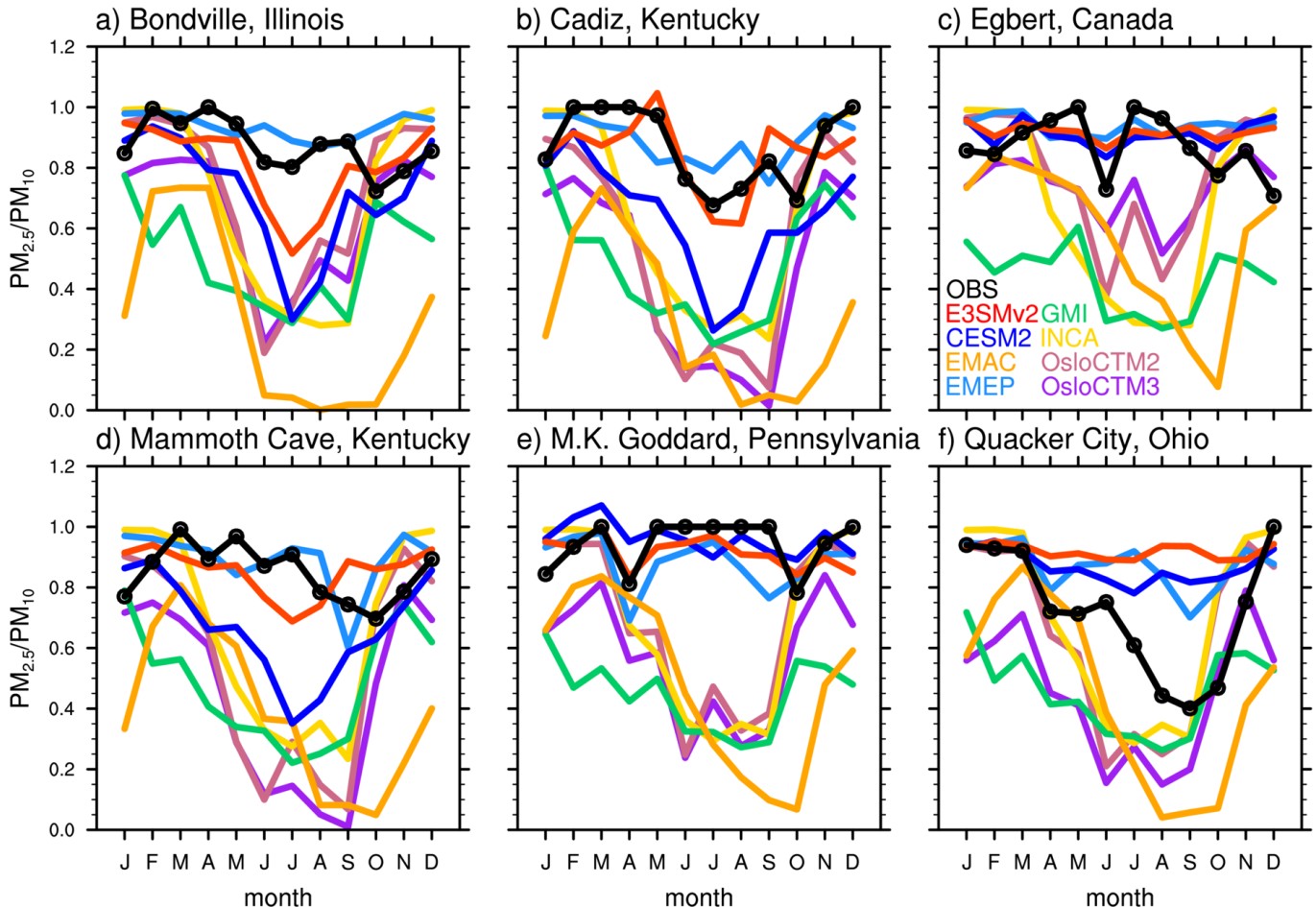

**Figure 8: Same as Figure 7 but for six co-located IMPROVE/CASTNET sites dominated by fine-mode nitrate: (a) Bondville (40.05ºN, 88.37ºW), (b) Cadiz (36.78ºN, 87.85ºW), (c) Egbert (44.23ºN, 79.78ºW), (d) Mammoth Cave (37.13ºN, 86.14ºW), (e) M.K. Goddard (41.43ºN, 80.15ºW), and (f) Quacker City (39.94ºN, 81.34ºW).**

Figure 9 compares the modeled vertical profiles of nitrate $PM_1/PM_4$ ratio from E3SMv2, CESM2, and EMEP to those derived from AMS ($PM_1$) and SAGA filters ($PM_4$) measurements during ARCTAS, DC3, SEAC[4]RS, WINTER, and KORUS-AQ campaigns. The observed nitrate $PM_1/PM_4$ ratios from ARCTAS are low (around or less than 0.2) near the surface (below 900 hPa), indicating that they are dominated by coarse $PM_{1-4}$ nitrate forming on sea salt and high-latitude dust. The observed peaks at 600 hPa in Fig 9a and at 900-850 hPa in Fig 9c are consistent with the peaks in Figs 4a and 4c, indicating that they are dominated by $PM_1$ fine nitrate likely from wildfire plumes. Observed $PM_1/PM_4$ ratios from DC3 and SEAC[4]RS are also low (around or less than 0.4) near surface (below 900 hPa), which is consistent with low surface $PM_{2.5}/PM_{10}$ ratios in Figs. 7a-7b and 7d-7f during late spring, summer, and early autumn. The coarse $PM_{1-4}$ nitrate forming on local dust and sea salt mainly contributes to the low $PM_1/PM_4$ ratios near the surface. Froyd et al. (2019) showed that mineral dust concentrations are high below 4 km in DC3 and SEAC[4]RS, which have considerable contribution to the total aerosol mass in DC3 (up to 50%). It is also found that dust and sea salt dominate aerosol

mass at low altitude over the Gulf of Mexico. In Figs. 9d-9e, observed $PM_1/PM_4$ ratios from DC3 increase with altitude from around 0.2 near the surface to larger than 0.8 at 500 hPa or above, indicating that fine $PM_1$ nitrate from pollution or wildfire becomes dominant in the middle troposphere. In Fig. 9f, the observed $PM_1/PM_4$ ratios from SEAC[4]RS are high (larger than 0.8) between 750 hPa and 550 hPa, which is consistent with the peak of $PM_1$ nitrate around 600 hPa caused by wildfire in the western U.S.

The observed $PM_1/PM_4$ ratios from WINTER are high (around or above 0.8) near the surface (below 900 hPa), which is consistent with the high surface $PM_{2.5}/PM_{10}$ ratios in Fig. 8 during February and March. The observed $PM_1/PM_4$ ratio from WINTER decreases with altitude from above 0.8 near the surface to less than 0.2 at/above 800 hPa, indicating that coarse $PM_{1-4}$ nitrate condensed on sea salt and dust particles may become dominant at/above the boundary layer. Similarly, the observed $PM_1/PM_4$ ratios from KORUS-AQ in Fig. 9j are relatively high near the surface but decrease with altitude to less than 0.2 around 600 hPa. Coarse $PM_{1-4}$ nitrate forming on dust and anthropogenic coarse PM may contribute to the observed low $PM_1/PM_4$ ratios above 800 hPa (Zhai et al., 2023). Vertical profiles from ground-based High Spectral Resolution Lidar (HSRL) show elevated dust layers transported from the Taklamakan and Gobi deserts during May 04-05, 2016 (Peterson et al., 2019). The HSRL vertical profiles also show high values of backscatter cross-section due to pollution/haze below 2.5 km during May 25-26. In general, there are large spreads in the modeled nitrate $PM_1/PM_4$ ratios among E3SMv2, CESM2 and EMEP. The modeled $PM_1/PM_4$ ratios can range from near-zero in EMEP to one in CESM2, as shown in Figs. 9a-9b (ARCTAS) and 9j-9k (KORUS-AQ). We can also see that the three models agree with observations and each other better in Fig. 9c (ARCTAS), Fig. 9e (DC3), and Fig. 9i (WINTER) than in other panels.

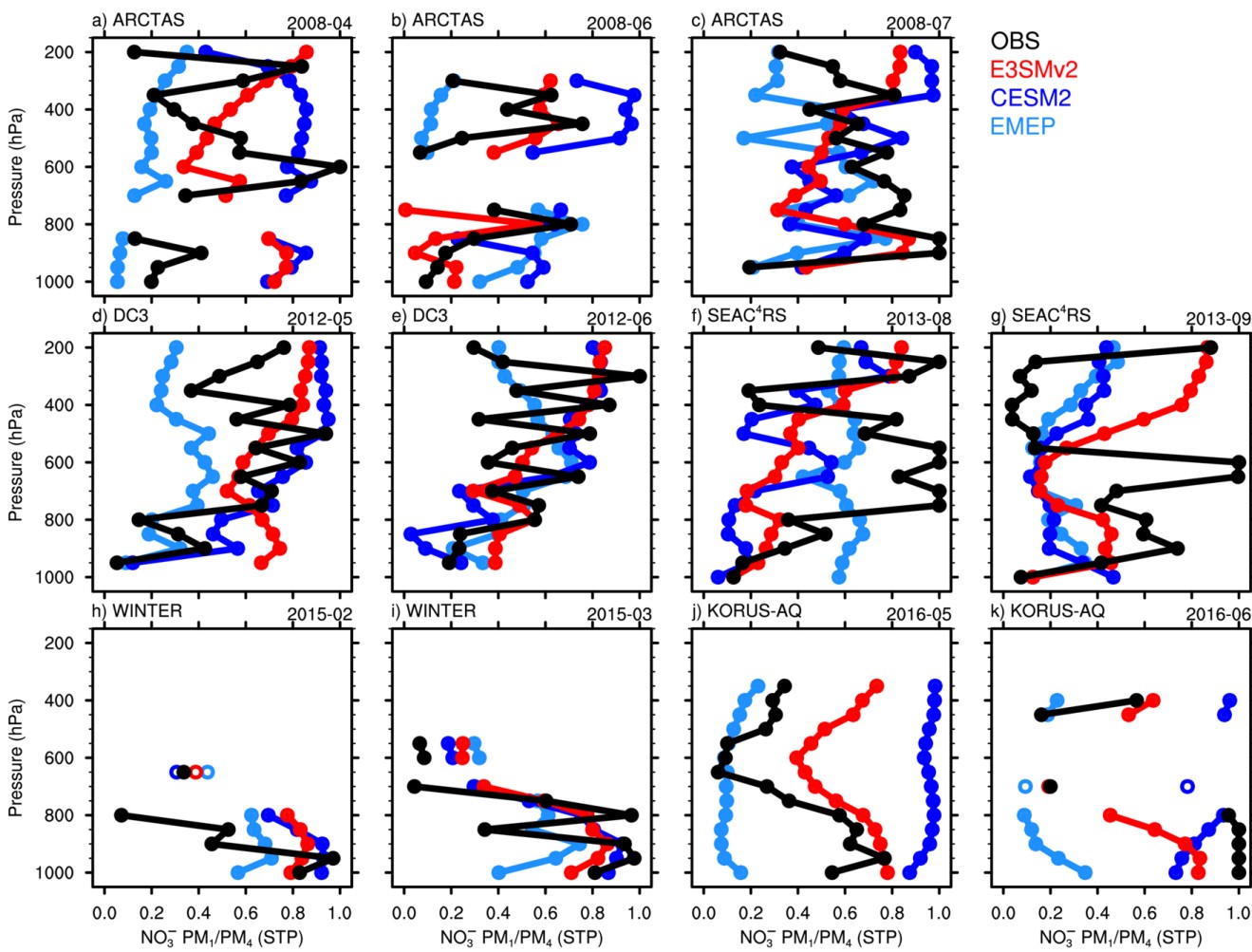

**Figure 9: Vertical profiles of nitrate PM$_1$/PM$_4$ ratios from model simulations (colored dots) and five aircraft campaigns: (a-c) ARCTAS, (d-e) DC3, (f-g) SEAC$^4$RS, (h-i) WINTER, and (j-k) KORUS-AQ (black dots).**

Figure 10 compares the modeled vertical profiles of nitrate PM$_1$/PM$_4$ ratio from E3SMv2, CESM2, and EMEP to those derived

from AMS (PM$_1$) and SAGA filters (PM$_4$) measurements during ATom flights. In general, observed PM$_1$/PM$_4$ ratios are low (less

than 0.4) below 600 hPa, despite the scarcity of measurements, due to the contribution of coarse PM$_{1-4}$ nitrate formed on sea salt.

Thompson et al. (2021) showed that large sea salt particles (diameter greater than 1 μm) dominate aerosol mass in the marine

boundary layer (MBL) (below 2 km) but have a significantly small contribution above MBL. The observed PM$_1$/PM$_4$ ratios tend

to be higher above 600 hPa than those below 600 hPa. Sometimes, fine-mode nitrate concentration measured by AMS (PM$_1$) is

455 larger than nitrate measured by SAGA filters (PM$_4$) above 600 hPa. Despite the measurement uncertainties, this may indicate that

fine PM$_1$ nitrate dominates the nitrate mass in the mid-to-upper troposphere. ATom field measurements show that small sulfate and

organic particles (diameter less than 1 μm) dominate aerosol mass above 2 km over the Southern Pacific and Atlantic and in the

upper troposphere (6-12 km) over tropical oceans, while considerable amount of coarse dust particles is present in the middle

troposphere (2-6 km) over tropical oceans and in the entire free troposphere over the North Pacific and Atlantic. E3SMv2 and CESM2 tend to better capture the vertical variations of $PM_1/PM_4$ ratio than EMEP that has much weaker vertical variations, likely due to biases in the model treatment of wet deposition and vertical transport. The three models tend to agree with each other better below 600 hPa than above 600 hPa.

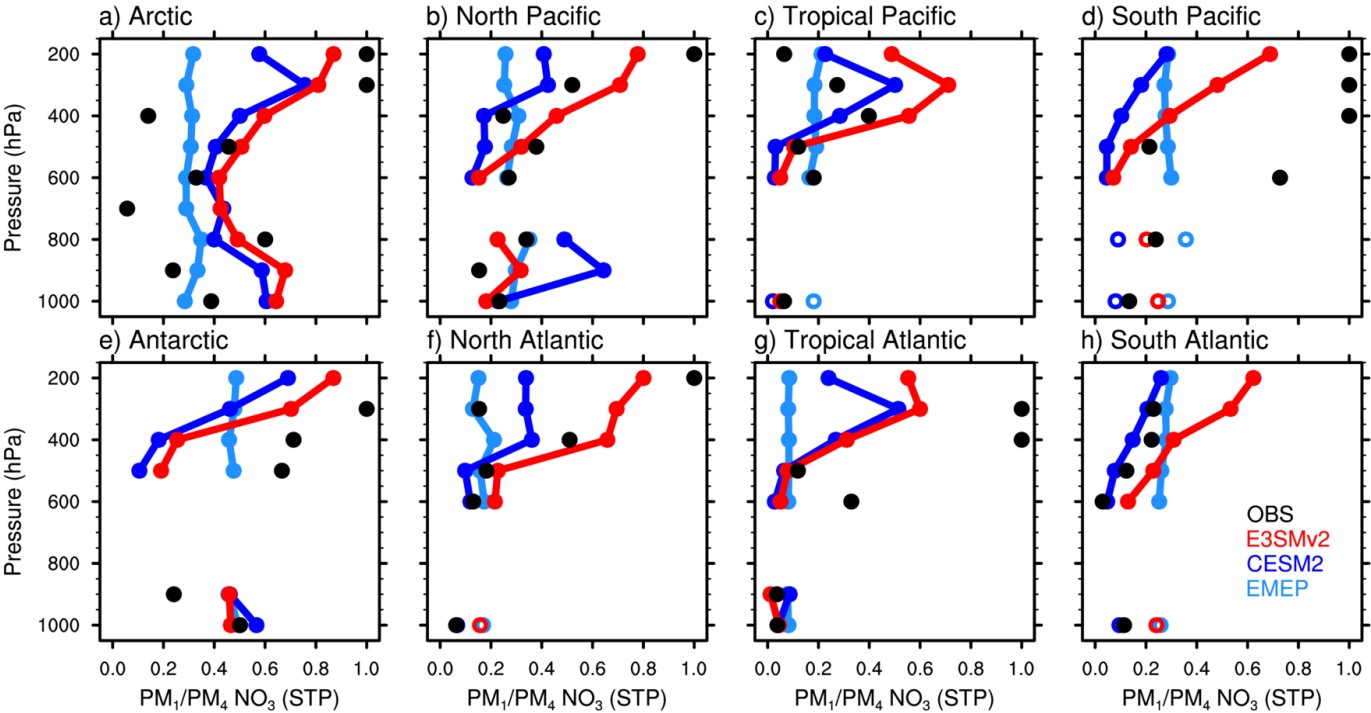

**Figure 10: Same as Figure 9 but for the eight regions (marked by boxes in Figure 2) during ATom 1-4 campaigns.**

### 3.3 Gas-aerosol Partitioning between Nitrate Aerosol and HNO₃

Figure 11 evaluates the modeled annual mean surface $NO_3^-/(NO_3^- + HNO_3)$ molar ratios from E3SMv2, CESM2, and AeroCom phase III models against ground-based observations from CASTNET sites over the U.S., EMEP sites over Europe, and EANET sites over East Asia. Overall, ground-based observations give an annual mean surface molar ratio of 0.52. The observed annual mean molar ratios at CASTNET, EMEP, and EANET sites are 0.46, 0.61, and 0.78, respectively. Observed surface molar ratios mostly fall between 0.2 and 0.8. In general, CESM2 and OsloCTM3 overestimate the surface molar ratio averaged over all sites, while the other AeroCom models underestimate the surface molar ratio. The simulated surface molar ratio in E3SMv2 agrees with the observations reasonably well. Note that the models' performance in simulating surface molar ratio does not necessarily align with global annual mean $NO_3^-/(NO_3^- + HNO_3)$ tropospheric burden ratio (Table 3). GMI produces the lowest surface molar ratio, which is consistent with the lowest global annual mean $NO_3^-/(NO_3^- + HNO_3)$ tropospheric burden ratio it produced among all

models (Table 3). However, EMEP underestimate the surface molar ratio, resulting in values lower than those of several other models, while producing the highest global annual mean $NO_3^-/(NO_3^- + HNO_3)$ tropospheric burden ratio among all models.

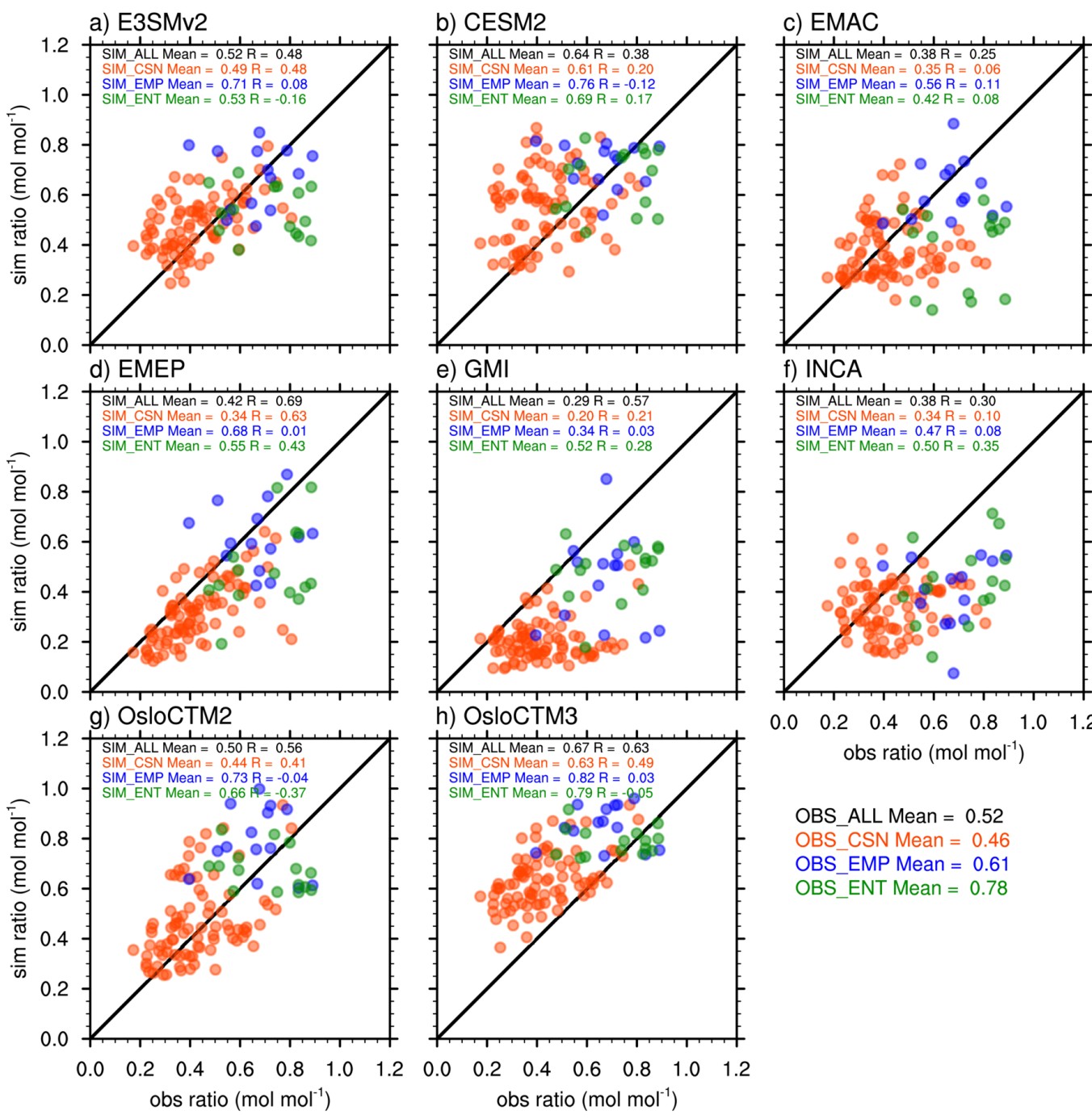

**Figure 11: Scatter plots of modeled annual mean surface $NO_3^-/(NO_3^- + HNO_3)$ molar ratios (mol mol$^{-1}$) compared with observations from CASTNET (CSN, red dots), EMEP (EMP, blue dots), and EANET (ENT, green dots). The numbers are mean ratios and correlation coefficients at the corresponding sites. Solid lines represent 1:1 comparison.**

Figures 12 and 13 show the seasonal variations of modeled surface $NO_3^-/(NO_3^- + HNO_3)$ molar ratio in comparison with observations at the same selected sites in Figs. 7 and 8, respectively. In general, there are large spreads in the modeled surface molar ratios. The large spreads in modeled HNO₃ surface concentrations (Figs. S7 and S8) partially contribute to the large uncertainties in the modeled surface molar ratios, which can be further related to model differences in multiple processes, such as gas-aerosol partitioning between nitrate aerosol and gas phase HNO₃, sulfate aerosol formation, gas phase chemistry (e.g., He et al., 2015), and wet removal of HNO₃ (e.g., Luo et al., 2019). Comparisons of nitrate surface concentration with CASTNET measurements are also provided in the supplement (Figs. S9 and S10). In Figure 12, the measured surface molar ratios at all six sites, which are dominated by coarse-mode nitrate, show relatively weak seasonal variations compared to Fig. 13. The measured molar ratios at Everglades NP and Virgin Island (Figs. 12a and 12b) are around 0.8 and 0.9, respectively, with small fluctuations. As shown in Figs. 7a and 7b, the two sites are dominated by coarse-mode nitrate formed through irreversible heterogeneous reactions on large dust and sea salt particles. The two sites also have weaker seasonal variations of sulfate aerosol than sites in the Northeastern U.S. (summer-high-winter-low) (see Figs. S3 and S4). These factors can contribute to the high molar ratios and weak seasonal variations at the two sites. OsloCTM2 and OsloCTM3 have positive biases in the modeled surface molar ratios at the two marine sites, which is mainly due to their gas-aerosol partitioning methods (only TEQMs). However, EMAC has negative biases at the two sites, which is mainly due to its significant positive biases in HNO₃. The observed relatively low molar ratios in summer at Acadia NP (Fig. 12c) is consistent with the relatively high sulfate concentrations (Fig. S3) and nitrate PM₂.₅/PM₁₀ ratios (Fig. 7c). The measured molar ratios at the three dusty sites (Figs. 12d-f) are between 0.2 and 0.6. Their seasonal variations are correlated with the seasonality of dust aerosol (see Fig. S2).

In Figure 13, the measured surface molar ratios at all six sites, which are dominated by fine-mode nitrate (mostly ammonium nitrate), show strong seasonal variations (summer-low-winter-high) due to seasonal variations of temperature, precipitation, and sulfate (Fig. S4), with maximum values between 0.5 and 0.8 during winter and minimum values around 0.2 during summer. All models capture the seasonal variations to varying degrees and have larger spread in the modeled molar ratio in winter, ranging from below 0.2 to above 0.8, than in summer. Although the model spread in the simulated HNO₃ at these sites is large across all seasons, the colder temperatures, lower precipitation, and lower sulfate concentrations at the same locations in winter promote more favorable conditions for nitrate formation compared to summer. Also, as indicated by Fig. 8, the nitrate formation pathway in winter is mainly through thermodynamic interactions between HNO₃ and NH₃ in most models, while nitrate formation through heterogeneous reactions on coarse dust and sea salt particles also tends to have a significant contribution in summer in many models. Unlike observations, many models produce the maximum of surface molar ratio in spring and autumn, especially E3SMv2. GMI consistently produces the lowest surface molar ratio across all months and significantly underestimates it during winter.

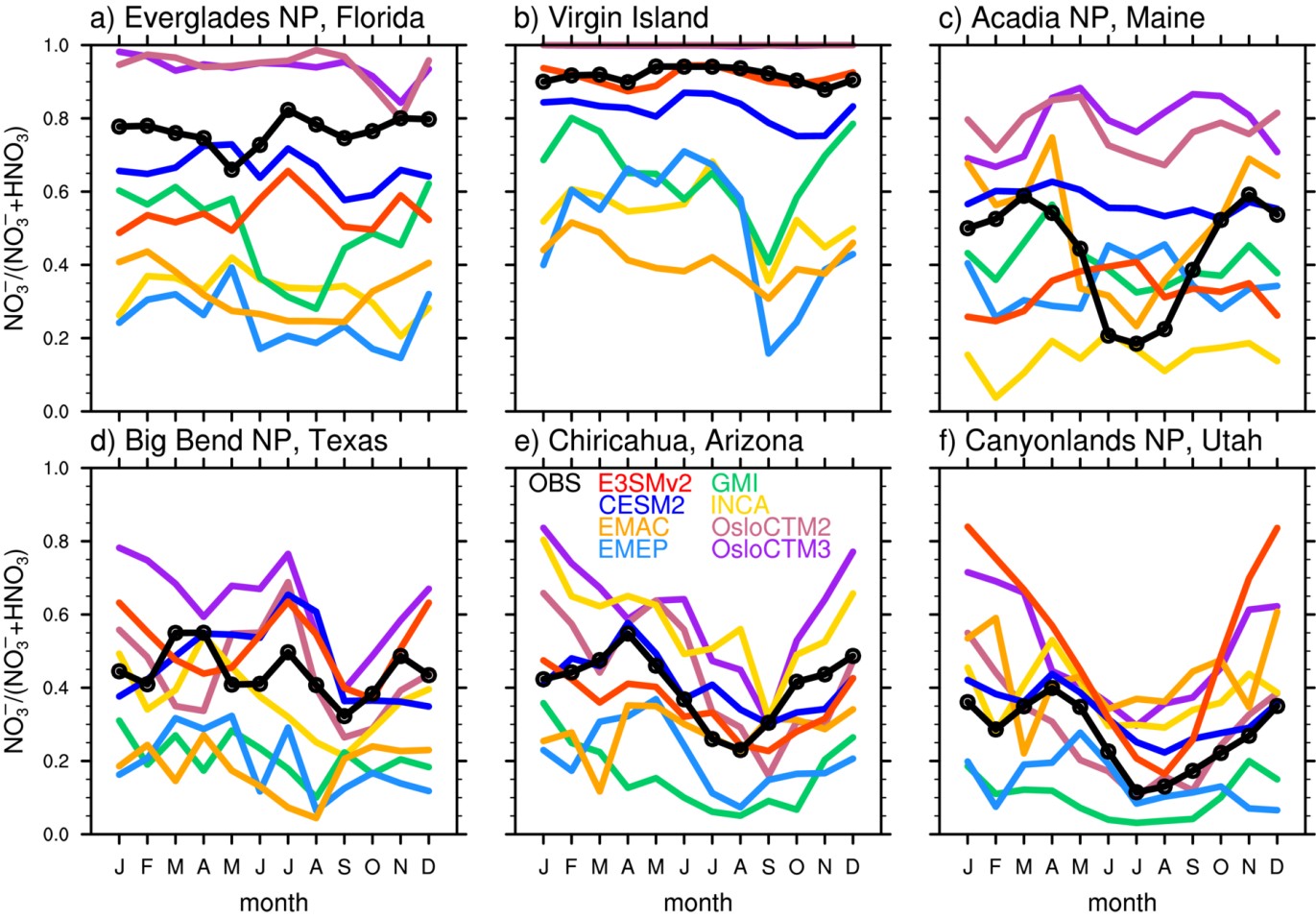

**Figure 12: Same as Figure 7 but for surface $NO_3^-/(NO_3^- + HNO_3)$ molar ratios (mol mol$^{-1}$).**

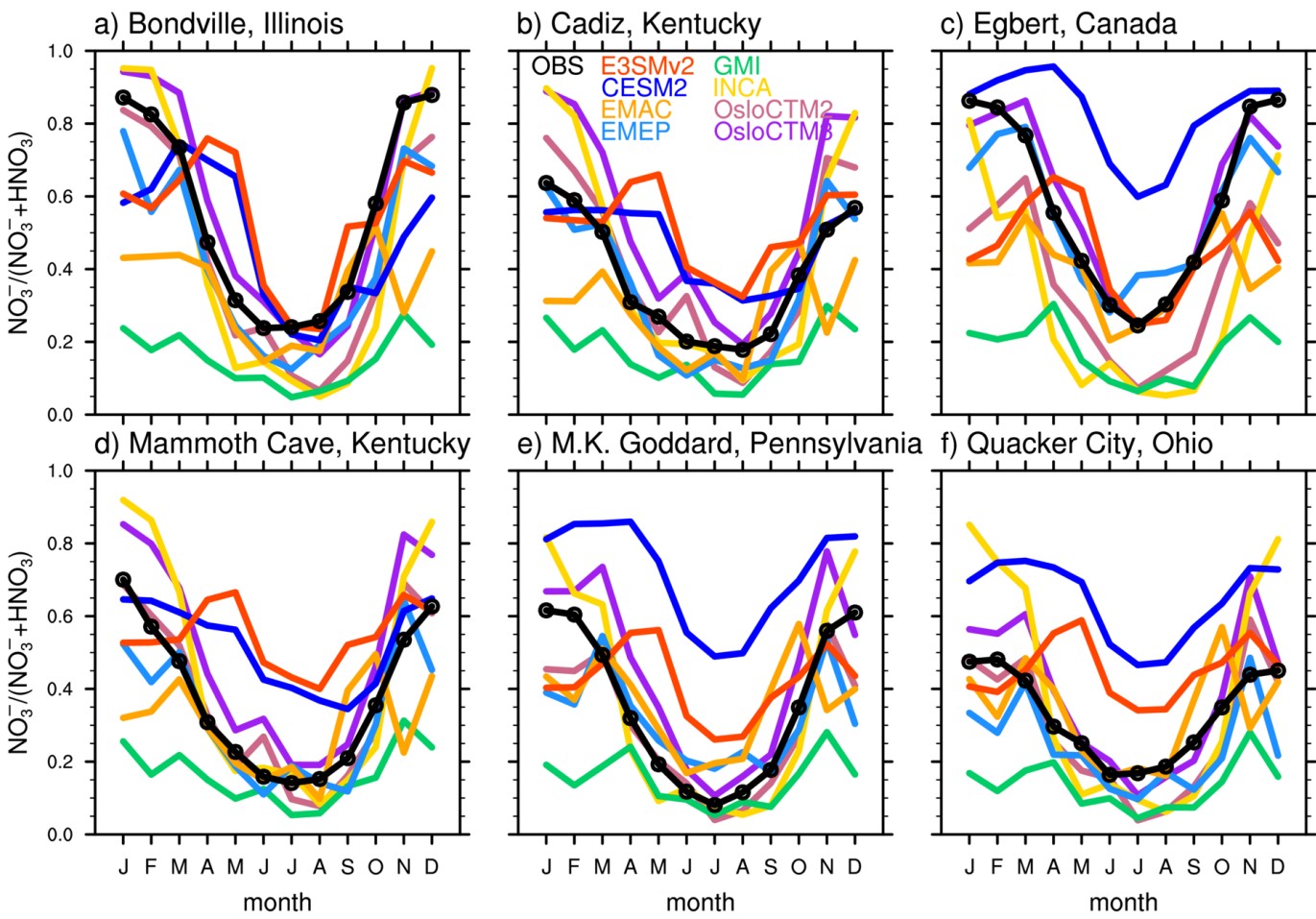

**Figure 13: Same as Figure 8 but for surface $NO_3^-/(NO_3^- + HNO_3)$ molar ratios (mol mol$^{-1}$).**

## 4 Discussion and Conclusions

In this study, we evaluate the simulated PM$_1$ and PM$_{2.5}$ nitrate concentrations, nitrate PM$_{2.5}$/PM$_{10}$ and PM$_1$/PM$_4$ ratios, and surface $NO_3^-/(NO_3^-+HNO_3)$ molar ratios from E3SMv2, CESM2, and six selected AeroCom phase III models against observations from multiple ground networks and aircraft campaigns. We find that both E3SMv2 and CESM2 overestimate the annual mean PM$_{2.5}$

nitrate surface concentration averaged over all IMPROVE, EMEP, ANTSO ASP, the South African, and the Japanese sites, whereas the six AeroCom models underestimate it. There are large spreads in the modeled vertical profiles of fine-mode nitrate, in some cases by two to three orders of magnitude. Most of the AeroCom models underestimate PM$_1$ nitrate concentrations below 600 hPa compared to the ARCTAS, DC3, SEAC$^4$RS, and KORUS-AQ campaigns. OsloCTM2 and OsloCTM3, which both use only TEQMs for the formation of coarse-mode nitrate, have uniformly significant negative biases in fine-mode nitrate concentration.

EMAC, which uses only TEQM but considers the kinetic limitation, has the smallest biases in the annual mean PM$_{2.5}$ nitrate surface

concentration among all models.

The observed nitrate PM$_{2.5}$/PM$_{10}$ and PM$_1$/PM$_4$ ratios are influenced by the relative contribution of fine sulfate/organic particles and coarse dust/sea salt particles. Overall, the ground-based observations give an annual mean surface nitrate PM$_{2.5}$/PM$_{10}$ ratio of 0.7 averaged across all sites with regional annual mean ratios at co-located IMPROVE/CASTNET, EMEP, the South African, and the Japanese sites of 0.82, 0.71, 0.56, and 0.57, respectively. Most models have negative biases in all four regions. The model differences in gas-aerosol partitioning and large uncertainties in the simulated lifecycle of dust and sea salt contribute to the uncertainties in simulating mass size distribution of nitrate. E3SMv2, EMEP, and INCA slightly overestimate the annual mean surface nitrate PM$_{2.5}$/PM$_{10}$ ratio, but they still have negative biases for considerable number of sites. EMAC, OsloCTM2, and OsloCTM3 which all use only TEQMs for the formation of coarse-mode nitrate have larger biases than the other three AeroCom models that use the first-order loss approximation. E3SMv2 (coupled with MOSAIC) tends to agree reasonably well with observations, while CESM2 (also coupled with MOSAIC) has negative biases in all four regions. The observed PM$_{2.5}$/PM$_{10}$ ratios at two U.S. marine sites are around 0.3 and 0.2, respectively, with small fluctuations, while the observed PM$_{2.5}$/PM$_{10}$ ratios at three dusty sites are higher but below 0.6 for most of the year. Quite a few of the models produce much stronger seasonal variations (summer-low-winter-high) of surface PM$_{2.5}$/PM$_{10}$ ratios than observations and have large negative biases during May to September at six selected sites over the U.S. where the observed surface nitrate PM$_{2.5}$/PM$_{10}$ ratios are around 0.8.

Observed PM$_1$/PM$_4$ ratios from ARCTAS, DC3, and SEAC[4]RS campaigns are low (around 0.2) near the surface (below 900 hPa), as they are dominated by coarse PM$_{1-4}$ nitrate formed on sea salt and dust. The observed PM$_1$/PM$_4$ ratios from WINTER and KORUS-AQ campaigns are high (around 0.8) near the surface, as they are dominated by fine PM$_1$ nitrate from anthropogenic pollution. The observed PM$_1$/PM$_4$ ratio increases with altitude from 0.2 to 0.8 in DC3 and SEAC[4]RS but decreases with altitude from 0.8 to 0.2 in WINTER and KORUS-AQ. In general, there are large spreads in the modeled PM$_1$/PM$_4$ ratios of nitrate from E3SMv2, CESM2 and EMEP. The modeled PM$_1$/PM$_4$ ratios can range from near zero in EMEP to one in CESM2 in ARCTAS and KORUS-AQ. The observed PM$_1$/PM$_4$ ratios from ATom are also low (less than 0.4) below 600 hPa, while fine-mode nitrate tends to dominate in the middle to upper troposphere. E3SMv2 and CESM2 tend to capture the observed strong vertical variations in PM$_1$/PM$_4$ ratios better than EMEP.

Overall, ground-based observations give an annual mean surface $NO_3^-/(NO_3^- + HNO_3)$ molar ratio of 0.52 averaged across all sites with regional annual mean ratios at CASTNET, EMEP, and EANET sites of 0.46, 0.61, and 0.78, respectively. In general, most models overestimate the surface molar ratio averaged over all sites. The observed surface molar ratios at sites dominated by coarse-mode nitrate show relatively weak seasonal variations. The measured molar ratios at the two marine sites are high (around or above 0.8) with small fluctuations, while the measured molar ratios at the three dusty sites are relatively low (between 0.2 and 0.6). The observed surface molar ratios at sites dominated by fine-mode nitrate show strong seasonal variations (summer-low-

winter-high) with maximum values between 0.5 and 0.8 during winter and minimum values around 0.2 during summer. There are large spreads in the modeled surface molar ratios at the selected sites, which is caused by not only the model differences in gas-aerosol partitioning but also the large uncertainties in modeled $HNO_3$ or total nitrate ($NO_3^- + HNO_3$). Multiple processes such as gas phase chemistry ($O_3$-$NO_x$-$HO_x$ chemistry or $N_2O_5$ hydrolysis) and wet removal of $HNO_3$ can also greatly affect the abundance

of $HNO_3$ and therefore change the molar ratio. In addition to those processes, sulfate aerosol formation as well as other processes can affect the abundance of free $NH_3$ available to react with $HNO_3$ and form particulate ammonium nitrate. All models capture the seasonal variations to varying degrees but have larger spread in the modeled molar ratio in winter than in summer at the sites dominated by fine-mode nitrate. Differences in temperature, precipitation and sulfate concentrations, which favor nitrate formation in winter, and different dominant nitrate formation pathways contribute to the different model spread between winter and summer.

Our study indicates the importance of gas-aerosol partition parameterization and simulation of dust and sea salt in correctly simulating mass size distribution of nitrate. Our analysis suggests that future studies and model development efforts should better represent heterogeneous reactions of nitrate formation on coarse dust and sea salt particles and use first-order loss approximation with TEQMs or dynamic mass transfer approach for gas-aerosol partitioning between nitrate aerosol and $HNO_3$ gas, as all models that use only TEQMs have larger biases than the other models. Using TEQMs only may also significantly underestimate fine-mode

nitrate compared with the other two methods. The large spread in modeled lifecycle of dust and sea salt can largely affect the mass size distribution of nitrate in GCMs and CTMs. Joint measurements of both fine- and coarse-mode nitrate (e.g., $PM_{2.5}$ and $PM_{10}$) with sulfate, ammonium, dust, sea salt, and related gases (e.g., $HNO_3$, $NH_3$, and $SO_2$) using a unified sampling protocol would greatly benefit future studies.

Although E3SMv2 and CESM2 have a lot of similarities in physical and chemical parameterizations, there are considerable

differences between the two models in the modeled $PM_1$ and $PM_{2.5}$ nitrate concentrations, nitrate $PM_{2.5}/PM_{10}$ and $PM_1/PM_4$ ratios, and surface $NO_3^-/(NO_3^-+HNO_3)$ molar ratios. Here we briefly discuss some important differences between the two models in aerosol parameterizations which can significantly impact the simulation of nitrate aerosol. Both E3SMv2 and CESM2 use the same dust emission scheme (Zender et al., 2003) but with different source function, which results in substantially different spatial distributions of dust emission (Wu et al. 2020). Compared to E3SMv2, CESM2 reduces the geometric standard deviation in the

accumulation and coarse modes of MAM4, which greatly reduces the dry deposition velocities for coarse dust and sea salt particles and increases the lifetime of dust and sea salt aerosol (Wu et al., 2020). Both E3SMv2 and CESM2 have the issue in the numerical coupling of aerosol emissions, dry deposition, and turbulent mixing. However, this issue has much more significant impact on the lifecycle of dust and sea salt in E3SMv2, as the lowest model level in E3SMv2 is much closer to the surface than the one in CESM2 (Wan et al., 2024). EAMv2 adopts most of the tunable parameters from the recalibrated atmosphere model, EAMv1p, which

significantly improves the simulations of clouds and precipitation climatology but increases anthropogenic aerosol load, such as

sulfate, and introduce larger biases in some aerosol-related fields, such as AOD (Ma et al., 2021; Golaz et al., 2022).

It should be noted that the comparisons between model results and observations are subject to considerable spatiotemporal representativeness errors. The selected AeroCom models were only run for 2008, while E3SMv2 and CESM2 were run for 2005-2014. All aircraft campaigns except for ARCTAS are outside the model simulation year (2008) of AeroCom phase III models. WINTER, KORUS-AQ, and ATom are not within the simulation period (2005-2014) of E3SMv2 and CESM2. Bian et al. (2017) showed that there are considerable differences between monthly and daily model results when compared with measurements from ARCTAS. All AeroCom models except EMEP have a relatively coarser horizontal resolution (2-3 degrees), compared to the 1-degree resolution of E3SMv2 and CESM2. The ATom deployments provided fewer data samples than the other five aircraft campaigns. It's also worth noting that differences in sampling protocols of ground networks between $PM_{2.5}$ and $PM_{10}$/TPM and aircraft campaigns between $PM_1$ and $PM_4$ make the evaluation of model results challenging.

*Code availability.* The E3SM-MOSAIC source code is available at https://github.com/E3SM-Project/E3SM/tree/mingxuanwupnnl/atm/trop_strat_mam4_mosaic_maint2.0.

*Data availability.* CASTNET data can be downloaded from https://www.epa.gov/castnet/download-data. IMPROVE data can be downloaded from https://vista.cira.colostate.edu/Improve/improve-data/. EMEP data can be downloaded from https://ebas-data.nilu.no/Default.aspx. EANET data can be downloaded from https://monitoring.eanet.asia/document/public/index. ARCTAS data are available at https://www-air.larc.nasa.gov/cgi-bin/ArcView/arctas. DC3 data are available at https://www-air.larc.nasa.gov/cgi-bin/ArcView/dc3. SEAC[4]RS data are available at https://www-air.larc.nasa.gov/cgi-bin/ArcView/seac4rs. WINTER data are available at https://data.eol.ucar.edu/master_lists/generated/winter/. KORUS-AQ data are available at https://www-air.larc.nasa.gov/missions/korus-aq/index.html. ATom data are available at https://espo.nasa.gov/atom. The AeroCom model output is archived at AeroCom server.

*Author contributions.* MW conceived the project, designed the experiments for E3SMv2 and CESM2, and ran the model simulations for E3SMv2 with input from HW. ZL ran the model simulations for CESM2 with input from XL. HB helped on the data processing of AeroCom model output. DC processed and provided the observation data at ANSTO ASP sites over Australia. MW performed the analysis with help and input from HW, HB, ZL, XL and YF. MW wrote the manuscript with contributions from all co-authors.

*Competing interests.* Some co-authors are members of the editorial board of Atmospheric Chemistry and Physics. The authors have no other competing interests to declare.

*Acknowledgement.* This research was supported as part of the Energy Exascale Earth System Model (E3SM) Science Focus Area, funded by the U.S. Department of Energy, Office of Science, Office of Biological and Environmental Research Earth System Model Development program area. The Pacific Northwest National Laboratory (PNNL) is operated for DOE by Battelle Memorial Institute under contract DE-AC05-76RLO1830. The work at the Lawrence Livermore National Laboratory (LLNL) was performed under the auspices of the U.S. DOE by LLNL under Contract DE-AC52-07NA27344. We would like to thank Syuichi Itahashi for providing measurements of $PM_{2.5}$ and $PM_{10}$ nitrate surface concentrations at Fukue site in Japan.

*Financial support.* This work is supported by the U.S. Department of Energy (DOE), Office of Science, Office of Biological and Environmental Research, Earth and Environmental System Modeling program as part of the Energy Exascale Earth System Model (E3SM) project.

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
