# Peer review of "Observationally Constrained Analysis on the Distribution of Fine and Coarse Mode Nitrate in Global Models"

_EGUsphere, 2025_

## Author Comment (AC1)

We thank the two anonymous reviewers for their valuable comments and constructive suggestions on the manuscript. Below, we explain how the comments and suggestions are addressed and highlight revisions made in the revised manuscript. The reviewers' comments are in blue color. Our responses are in black, and our corresponding revisions in the manuscript are in red.

**Reviewer #1**

**General Comments:**
Particulate nitrate plays a critical role in aerosol radiative forcing and leads to very large uncertainty in climate projection. But very few global climate models have nitrate included, and only sparse studies evaluated the nitrate simulation performance. This work provides a comprehensive evaluation of the nitrate simulation in six global climate models again in-situ observations over north American, Australia, Japan and south Asia, Europe and Africa. This work not only evaluate the simulation of nitrate mass concentration at the surface level, but also for the vertical profile and gas-aerosol partitioning of nitrate. I think this work fits well with the scope of ACP. Although the scientific value is appreciated for this work, I would like to make a few suggestions hopeful could help improve the presentation of this article for publishing in ACP.

**Reply:** We thank the reviewer for the encouraging comments.

**Minor Comments:**
1. IPCC AR5 2013 is referred. I recommend refer to the latest IPCC AR6 2021, to keep the research up to date.

**Reply:** Thank you for the suggestion. We did cite IPCC AR6 chapter 6 (i.e., Szopa et al., 2021) in the first paragraph for the important role that nitrate plays in the Earth's climate and air quality. Unfortunately, IPCC AR6 does not have an updated estimate of nitrate radiative forcing. In Figure 6.12 from AR6 chapter 6 (Szopa et al., 2021), nitrate is not included in the breakdown of individual contributions by aerosol species to the total ERF and the contribution of $NO_x$ to the total ERF does not include nitrate either.

2. Line 75-77. This sentence is not clear. Please rephrase it.

**Reply:** We have rewritten two paragraphs in the introduction in response to a separate comment. The following revision addresses this comment:

"Previous studies found large relative differences (up to 150%) of nitrate concentrations at co-located desert and marine sites over the U.S. between the Clean Air Status and Trends Network (CASTNET) and the Interagency Monitoring of Protected Visual Environments (IMPROVE) (Ames and Malm, 2001; Sickles and Shadwick, 2008), which suggest significant fractions of coarse-mode nitrate ($PM_{>2.5}$, PM with diameter larger than 2.5 μm) related to heterogeneous reactions on dust and sea salt particles."

3. Thanks for providing comprehensive review of previous studies in nitrate simulation, in the Introduction section. However, the presentation, or the writing style, would need some improvement. Rather than just listing what has been done previously (e.g., L80-90, and many other), but also better to summarize a key message you would like to express to audience.

**Reply:** We have rewritten the two paragraphs in the introduction section (L63-L89) to deliver clear key messages to readers:

[revised manuscript text omitted]

4. "MOSAIC explicit treats the heterogeneous reactions on HNO3 on dust and sea salt particles …". What about particles of other species? Are reactions treated independently for each size mode, or a bulk treatment for particles spanning all modes?

**Reply:** In the MAM4 scheme, aerosol species are assumed to be internally mixed within modes and externally mixed among modes. Reactions are treated independently for each size mode. We revised the description in section 2.1.1 to clarify it.

"In MAM4, aerosol species are assumed to be internally mixed within modes and externally mixed among modes. Wu et al. (2022) added nitrate ($NO_3^-$) and ammonium ($NH_4^+$) aerosol to the Aitken, accumulation, and coarse modes of MAM4. MOSAIC explicitly and independently treats the heterogeneous reactions of $HNO_3$ on particles containing dust (i.e., $CaCO_3$) and/or sea salt (i.e., NaCl) in the Aitken, accumulation, and coarse modes."

5. Line-150. Why? Or you may want citation here to support your argument.

**Reply:** The equilibrium timescale for supermicron particles exceeds the typical GCM timestep, which is usually less than one hour. TEQMs does not account for the kinetic limitation of coarse particles and, therefore, may lead to an overestimation of nitrate in the coarse mode.

We have rewritten that part to address this comment:

"Previous studies showed that supermicron (coarse) nitrate particles take significantly longer, ranging from several hours to days, to reach equilibrium between the gas ($HNO_3$) and particles phase than submicron (fine) particles that equilibrate within minutes (Meng and Seinfeld, 1996; Fridlind and Jacobson, 2000). This equilibrium timescale for supermicron particles exceeds the typical GCM timestep, which is usually less than one hour. TEQMs assume instantaneous equilibrium between gas and particle phases with each timestep, which does not account for the kinetic limitation of coarse particles and therefore may lead to an overestimation of nitrate in the coarse mode."

6. last line of page-6. Change to "with 1 year spin-up and the last 10-year results for analysis".

**Reply:** Revised.

7. Line 184. Should be Table-2?

**Reply:** Corrected.

8. In the discussion, where possible, better to briefly engage some discussion about ammonia, sulfate, sea-salt and dust simulations. Since, this could help point towards clearer directions for improving nitrate simulation. Excessive ammonia after neutralizes sulfuric acid can then contribute to particulate nitrate.

**Reply:** We have added the following discussion in the conclusion:

"Most models have negative biases in all four regions. The model differences in gas-aerosol partitioning and large uncertainties in the simulated lifecycle of dust and sea salt contribute to the uncertainties in simulating mass size distribution of nitrate. E3SMv2, EMEP, and INCA slightly overestimate the annual mean surface nitrate $PM_{2.5}/PM_{10}$ ratio, but they still have negative biases for considerable number of sites."

"The observed surface molar ratios at sites dominated by fine-mode nitrate show strong seasonal variations (summer-low-winter-high) with maximum values between 0.5 and 0.8 during winter and minimum values around 0.2 during summer. There are large spreads in the modeled surface molar ratios at the selected sites, which is caused by not only the model differences in gas-aerosol partitioning but also the large uncertainties in modeled $HNO_3$ or total nitrate ($NO_3^- + HNO_3$). Multiple processes such as gas phase chemistry ($O_3$-$NO_x$-$HO_x$ chemistry or $N_2O_5$ hydrolysis) and wet removal of $HNO_3$ can also greatly affect the abundance of $HNO_3$ and therefore change the molar ratio. In addition to those processes, sulfate aerosol formation as well as other processes can affect the abundance of free $NH_3$ available to react with $HNO_3$ and form particulate ammonium nitrate."

"All models capture the seasonal variations to varying degrees but have larger spread in the modeled molar ratio in winter than in summer at the sites dominated by fine-mode nitrate. Differences in temperature, precipitation and sulfate concentrations, which favor nitrate formation in winter, and different dominant nitrate formation pathways contribute to the different model spread between winter and summer."

**Reviewer #2**

**General Comments:**
This manuscript focuses on characterizing the ability of atmospheric models to capture the mass size distribution of nitrate aerosols, by comparing a set of models with relevant observations from ground-based networks and aircraft campaigns. Simulating nitrate aerosols remains challenging, and this work is a valuable contribution for the modeling community in the direction of better constraining the representation of nitrate. The paper is within the scope of ACP, well-structured and written, and I recommend publication in ACP. I would like to offer some recommendations to improve the clarity and presentation of this article for publication:

**Reply:** We thank the reviewer for the encouraging comments.

**Major Comments:**
Section: 2.1 Model description
1. The E3SMv2 and CESM2 models couple MOSAIC with MAM4, however, when reading, it is not clear which nitrate relevant processes are treated by MOSAIC and which by MAM4. Can you add more detail?

**Reply:** The implementation of MOSAIC in E3SMv2 and CESM2 replaces the default MAM4 treatment of gas-aerosol exchange between gases, including $H_2SO_4$, $HNO_3$, $NH_3$, HCl, and a single lumped secondary organic aerosol (SOA) precursor, and aerosols.

To improve the clarity, we have now added the following description in Section 2.1.1 to give more details.

"Wu et al. (2022) implemented the Model for Simulating Aerosol Interactions and Chemistry (MOSAIC) module (Zaveri et al., 2008) in E3SMv2 and coupled it with the Model for Ozone and Related chemical Tracers gas chemistry (MOZART-4) (Emmons et al., 2010; Tilmes et al., 2015) and an enhanced version of the four-mode version of Modal Aerosol Module (MAM4) (Liu et al., 2016; Wang et al., 2020) following previous model development effort in CESM2 (Lu et al., 2021; Zaveri et al., 2021). MOSAIC is a comprehensive aerosol chemistry module which simulates the dynamic partitioning between semivolatile gases and particles of different sizes in an accurate but computationally efficient way. MOSAIC implemented in EAMv2 replaces the default MAM4 treatment of gas-aerosol exchange between gases, including $H_2SO_4$, $HNO_3$, $NH_3$, HCl, and a single lumped secondary organic aerosol (SOA) precursor, and aerosols. The aqueous chemistry (i.e., occurring with cloud water) is also modified to include reactions of $HNO_3$, $NH_3$, and HCl."

2. Is the size bins treatment in MOSAIC the same in E3SMv2 and CESM2, in terms of number and size range of bins for nitrate? I would suggest to explicitly add these information in the models description, since (mass) size distribution is a key focus of the paper. Also adding some additional information on how nitrate size distribution is treated in the other AeroCom models would be beneficial (maybe as a supplementary Table). This is important context also for the cutoff assumptions when comparing with observations.

**Reply:** The geometric standard deviations ($\sigma_g$) of particle sizes in the accumulation and coarse mode of MAM4 are different between CAM6 and EAMv2, which has significant impacts on the lifecycle of dust through dry deposition (Wu et al., 2020). The upper and lower bound of the number median diameter in the three modes are also different between CAM6 and EAMv2.

We have added details on the model treatment of aerosol size bins or modes containing nitrate to Table 1 for E3SMv2, CESM2, and the AeroCom models.

**Table 1: Nitrate chemical mechanisms and physical properties.**

| Model | Gas-aerosol partitioning method for ammonium nitrate (fine and coarse) | Gas chemistry | DU/SS-nitrate treatment (fine and coarse) | Bins/modes for nitrate | Resolution |
|---|---|---|---|---|---|
| E3SMv2 | DYN (MOSAIC) (Zaveri et al., 2008; Wu et al., 2022) | MOZART-4 (Emmons et al. 2010) | DYN | Aitken, accumulation, and coarse modes ($\sigma_g$=1.6, 1.8, and 1.8) | 1°, 72 |
| CESM2 | DYN (MOSAIC) (Zaveri et al., 2008, 2021; Lu et al., 2021) | MOZART-TS1 (Emmons et al. 2020) | DYN | Aitken, accumulation, and coarse modes ($\sigma_g$=1.6, 1.6, and 1.2) | 1.25°×0.9°, 56 |
| EMAC | TEQM (ISORROPIA-II) (Fountoukis and Nenes, 2007) | MECCA (Sander et al., 2011) | TEQM | nucleation, Aitken, accumulation, and coarse modes | 2.8°×2.8°, 31 |
| EMEP | TEQM (MARS) (Saxena et al., 1986) | EmChem09 (Simpson et al., 2012) | first-order loss | fine and coarse modes ($D_p$=0.33 and 3.0 μm; $\sigma_g$=1.8 and 1.6) | 0.5°×0.5°, 20 |
| GMI | TEQM (RPMARES) (Saxena et al., 1986) | Strahan et al., 2007 | first-order loss | three bins ($D_p$<0.1, 0.1-2.5, >2.5 μm) | 2.5°×2°, 72 |
| INCA | TEQM (INCA) (Hauglustaine et al., 2004) | Hauglustaine et al., 2004; Folberth et al., 2006 | first-order loss | fine and coarse modes | 1.9°×3.75°, 39 |
| OsloCTM2 | TEQM (EQSAM_v03d) (Metzger and Leliveld, 2007) | Berntsen and Isaksen, 1997 | TEQM (only SS) | fine ($D_p$=0.1 μm; $\sigma_g$=2.0) and coarse modes | 2.8°×2.8°, 60 |

| OsloCTM3 | TEQM (EQSAM_v03d) | Berntsen and Isaksen, 1997 | TEQM (only SS) | fine ($D_p$=0.1 μm; $\sigma_g$=2.0) and coarse modes | 2.25°×2.25°, 60 |

Note. DYN: dynamic mass transfer; DU: dust; SS: sea salt.

We have also added the following discussion:

"Nitrate and ammonium aerosol are explicitly simulated in the Aitken, accumulation, and coarse modes. However, as shown in Table 1, the geometric standard deviations ($\sigma_g$) in the accumulation and coarse mode of MAM4 are different between CAM6 and EAMv2, which has significant impacts on the lifecycle of dust through dry deposition (Wu et al., 2020). The upper and lower bound of the number median diameter in the three modes are also different between CAM6 and EAMv2."

"AeroCom models have different treatments for the size distribution of nitrate, which affects the calculation of nitrate concentration at cut off size."

Section 2.3 Observations
3. Ground based observations: are the measurements of nitrate assumed to be dry mass (RH=0%) or some level of hydration is taken into account in the measured mass? In particular, are these assumptions the same for CASTNET and IMPROVE, and if not, which source of error can this introduce in the PM2.5/PM10 mass ratio, given that nitrate is strongly hydrophilic?

**Reply:** The measurements of nitrate are reported as dry mass. We believe that the water vapor should have limited impact on the measurements of nitrate. Both CASTNET and IMPROVE use ion chromatography (IC) to analyze the extract from the Teflon and nylon filter, respectively, instead of measuring gravimetric mass of the material collected on the filters (before and after sampling). The material collected on the filters is extracted by extraction solution and subsequently analyzed for nitrate anions by a Dionex IC. We have discussed the main differences between CASTNET and IMPROVE sampling protocols and their impacts on biases in estimating $PM_{2.5}$/$PM_{10}$ ratios in section 2.3.

Aircraft observations: as acknowledged, not all the simulated periods in the models overlap with of aircraft campaigns deployment, thus comparison need to be treated cautiously. Can you briefly add which sources of uncertainty this might introduce and are they stronger at different heights (e.g. above/below PBL)?

**Reply:** The differences in the anthropogenic, biomass burning, and dust emissions between the simulation period and the observation period may contribute to the uncertainty of the comparison. Those differences in emissions may have large impact on

the simulated mass size distribution of nitrate in the middle to upper troposphere over remote oceans, regions sampled during the ATom campaigns.

We have added a few words to clarify this.

"The model biases in fine-mode nitrate and nitrate $PM_1/PM_4$ ratio, when compared to aircraft measurements, should be interpreted with caution due to discrepancies, particularly in the anthropogenic, biomass burning, and dust aerosol emissions, between the simulation period and the observation period. Those differences in emissions may have stronger impact on the simulated mass size distribution of nitrate in the middle to upper troposphere than in the lower troposphere over remote oceans (e.g., ATom), as sea salt is dominant in the marine boundary layer and fine sulfate/carbonaceous aerosols and coarse dust aerosols are dominant in the middle to upper troposphere (Thompson et al., 2021)."

Section 3 Results
5. Figures 4, 5, 9, 10: For added clarity in the vertical distribution plots, I suggest to plot observations on top of all the model lines, as done for the surface concentrations plots (e.g. fig 7).

**Reply:** Following the great suggestion, we have revised the figures to plot observations on top of all model lines to improve the clarity.

[Figure]

**Figure 4: Vertical profiles of fine-mode nitrate concentrations (μg m⁻³ in STP) from model simulations (colored lines) and PM₁ nitrate concentrations from five aircraft campaigns: (a-c) ARCTAS, (d-e) DC3, (f-g) SEAC⁴RS, (h-i) WINTER, and (j-k) KORUS-AQ. Black dots denote mean values of observations with one standard deviation on each side marked in grey lines. The numbers in each panel are median concentrations.**

[Figure]

**Figure 5: Vertical profiles of fine-mode nitrate concentrations (μg m⁻³ in STP) from model simulations (colored dots) and PM$_1$ nitrate concentrations from ATom 1-4 campaigns (with black dots for mean values and grey lines marking one standard deviation of observations). The numbers in each panel are median concentrations.**

[Figure]

**Figure 9: Vertical profiles of nitrate PM$_1$/PM$_4$ ratios from model simulations (colored dots) and five aircraft campaigns: (a-c) ARCTAS, (d-e) DC3, (f-g) SEAC$^4$RS, (h-i) WINTER, and (j-k) KORUS-AQ (black dots).**

[Figure]

**Figure 10: Same as Figure 9 but for the eight regions (marked by boxes in Figure 2) during ATom 1-4 campaigns.**

Section 4 Discussion and Conclusion
6. L481-517: This first part of the discussion just states the different results of the model observation comparison but doesn't explicitly links them to the uncertainty in the representation of underlying processes. E.g. L516: All models capture the seasonal variations to varying degrees but have larger spread in the modeled molar ratio in winter. Why? Which processes are causing the model spread particularly in winter? This part would benefit from an additional discussion on the potential sources of uncertainty in the modeled processes that may be driving the observed biases in the comparison.

**Reply:** Further data analysis has revealed large spreads in modeled HNO$_3$ surface concentrations. The large spreads in the modeled surface molar ratios at the selected sites is caused by not only the model differences in in gas-aerosol partitioning but also the large uncertainties in modeled HNO$_3$ or total nitrate (NO$_3^-$ + HNO$_3$). Multiple processes such as gas phase chemistry (O$_3$-NO$_x$-HO$_x$ chemistry or N$_2$O$_5$ hydrolysis) and wet removal of HNO$_3$ can also greatly affect the abundance of HNO$_3$ and therefore change the molar ratio. In addition to those processes, sulfate aerosol formation as well as other processes can affect the abundance of free NH$_3$ available to react with HNO$_3$ and form

particulate ammonium nitrate.

[revised manuscript text omitted]

For conciseness, we have put the related figures in supplement.

[Figure]

**Figure S7.** Same as Figure 7 but for HNO₃ surface concentrations (μg m⁻³).

[Figure]

**Figure S8.** Same as Figure 8 but for $HNO_3$ surface concentrations ($\mu g\ m^{-3}$).

[Figure]

**Figure S9.** Same as Figure 7 but for nitrate surface concentrations ($\mu g\ m^{-3}$).

[Figure]

**Figure S10.** Same as Figure 8 but for nitrate surface concentrations (μg m$^{-3}$).

7. L521-523: the results suggest that coarse mode nitrate is better captured when representing heterogenous formation on sea salt and dust as first order loss approximation or dynamic mass transfer. I know the models setup is very different among models, but can you derive any conclusion from the results in terms of which method seems to be better capturing fine mode nitrate (simple TEQM, more complex TEQM, dynamic mass transfer)?

**Reply:** It is difficult to derive any conclusion in terms of which gas-aerosol partitioning method (TEQM only, TEQM combined with first order loss approximation, and dynamic mass transfer) lead to smaller model biases in simulating fine-mode nitrate.

In Figure 3, OsloCTM2 and OsloCTM3, which use only TEQMs, have similar negative biases in the annual mean PM$_{2.5}$ nitrate surface concentration averaged across all sites as GMI and INCA, which use TEQMs with first order loss approximation. EMAC, which uses only TEQM but considers the kinetic limitation, has the smallest biases in the annual mean PM$_{2.5}$ nitrate surface concentration among all models. OsloCTM2 and OsloCTM3 have much larger negative biases than other models at the Australian, South African, and Japanese sites, which is consistent with that the two models have much larger negative

biases in $PM_{2.5}/PM_{10}$ at the South African and Japanese sites.

In Figures 4 and 5, OsloCTM2 and OsloCTM3 give much lower fine-mode nitrate than other models and have significant negative biases compared with all aircraft campaigns. However, some models (e.g., INCA and EMAC) have much larger biases (positive) than OsloCTM2 and OsloCTM3.

We have also revised the text in the conclusion section to add implications on simulating fine-mode nitrate:

"Most of the AeroCom models underestimate $PM_1$ nitrate concentrations below 600 hPa compared to the ARCTAS, DC3, SEAC[4]RS, and KORUS-AQ campaigns. OsloCTM2 and OsloCTM3, which both use only TEQMs for the formation of coarse-mode nitrate, have uniformly significant negative biases in fine-mode nitrate concentration. EMAC, which uses only TEQM but considers the kinetic limitation, has the smallest biases in the annual mean $PM_{2.5}$ nitrate surface concentration among all models."

"Our analysis suggests that future studies and model development efforts should better represent heterogeneous reactions of nitrate formation on coarse dust and sea salt particles and use first-order loss approximation with TEQMs or dynamic mass transfer approach for gas-aerosol partitioning between nitrate aerosol and $HNO_3$ gas, as all models that use only TEQMs have larger biases than the other models. Using TEQMs only may also significantly underestimate fine-mode nitrate compared with the other two methods."

**Minor Comments:**
1. L86-89: "Zhai et al (2023)...found that including anthropogenic coarse particulate matter mainly composed of anthropogenic dust, significantly reduces the overestimation of fine-mode nitrate in previous versions of GEOS-Chem" add what process was responsible for this decrease (less HNO3 available for fine mode nitrate?)

**Reply:** Including nitrate formation through heterogeneous reactions on coarse dust particles competes for $HNO_3$ with formation of ammonium nitrate on fine particles and therefore improves the agreement with observations. We have rewritten this part to address this comment:

"Previous regional modeling studies show that including nitrate formation on coarse sea salt and dust particles through heterogenous reactions can significantly shift the mass size distribution of nitrate aerosol (e.g., Chen et al., 2020; Zhai et al., 2023), as it competes for $HNO_3$ with formation of ammonium nitrate on fine particles. Chen et al. (2020) showed

that heterogeneous reactions on sea salt shift mass size distribution of nitrate from fine to coarse mode compared with an experiment turning off sea salt emission using WRF-Chem. The simulated mass size distribution of nitrate agrees well with measurements at Melpitz in Europe, where coarse-mode nitrate ($PM_{>1.2}$, PM with diameter larger than 1.2 μm) accounts for ~20% of total nitrate aerosol in marine air mass in September. Zhai et al. (2023) compared simulated nitrate concentrations from GEOS-Chem with observations ($PM_1$ and $PM_4$, PM with diameter less than 1 and 4 μm, respectively) from the Korea-United States Air Quality (KORUS-AQ) campaign and found that including heterogeneous reactions on anthropogenic coarse particulate matter, mainly composed of anthropogenic dust, significantly reduces the overestimation of fine-mode nitrate in previous versions of GEOS-Chem."

2. Table 1: I would add for additional clarity which nitrate mode (fine/coarse) is treated in by the 'Gas-aerosol partitioning method'(fine only) and the 'DU/SS-nitrate treatment' (fine+coarse).

**Reply:** We have changed "Gas-aerosol partitioning method" to "Gas-aerosol partitioning method for ammonium nitrate (fine and coarse)" and "DU/SS-nitrate treatment" to "DU/SS-nitrate treatment (fine and coarse)" in Table 1 for additional clarity.

3. L22-225: for E3SMv2 and CESM2, are monthly output from 2014-only the one chosen for comparison to WINTER, KORUS-AQ and Atom? Please specify the sampling year of model output.

**Reply:** We have added a few words to clarify it.

"We use the corresponding monthly mean model results for the aircraft campaigns operated during the simulation period. Note that all aircraft campaigns except for ARCTAS are outside the model simulation year (2008) of AeroCom phase III models. For comparisons with WINTER, KORUS-AQ, and ATom campaigns that are not within the simulation period (2005-2014) of E3SMv2 and CESM2, we use the ten-year average of the corresponding month from the two models."

4. Table 3: Is there a reason why you have chosen to report the data in terms of moles instead of mass? For most studies (like in Bian et al.2017) they are expressed in terms of mass (Tg, mass mixing ratio).

**Reply:** We are not reporting the data of nitrate and $HNO_3$ in mol. We only report the

tropospheric burden ratio of $NO_3^-/(NO_3^- + HNO_3)$ and the burden ratio of $PM_{2.5}/PM_{10}$ nitrate in mol mol$^{-1}$ which is essentially unitless and can better represent the partitioning of between reactive nitrogen species. We report $PM_{2.5}$ nitrate, total nitrate, and tropospheric $HNO_3$ burden in mass of N (Tg N) which is widely used in literature and comparable to other studies (e.g., Hauglustaine et al., 2014; Zaveri et al., 2021).